# Signatures of optimal codon usage in metabolic genes inform budding yeast ecology

**Abigail Leavitt LaBella**[1], **Dana A. Opulente**[2], **Jacob L. Steenwyk**[1], **Chris Todd Hittinger**[3], **Antonis Rokas**[1]*

**1** Department of Biological Sciences, Vanderbilt University, Nashville, Tennessee, United States of America,
**2** Department of Biology, Villanova University, Villanova, Pennsylvania, United States of America,
**3** Laboratory of Genetics, DOE Great Lakes Bioenergy Research Center, Wisconsin Energy Institute, Center for Genomic Science Innovation, J.F. Crow Institute for the Study of Evolution, University of Wisconsin-Madison, Madison, Wisconsin, United States of America

* antonis.rokas@vanderbilt.edu

**Data Availability Statement:** All analyses were done on publicly available and published genome assemblies and annotations (where available). The codon optimization values were obtained from the figshare repository from LaBella et al. 2019

## Abstract

Reverse ecology is the inference of ecological information from patterns of genomic variation. One rich, heretofore underutilized, source of ecologically relevant genomic information is codon optimality or adaptation. Bias toward codons that match the tRNA pool is robustly associated with high gene expression in diverse organisms, suggesting that codon optimization could be used in a reverse ecology framework to identify highly expressed, ecologically relevant genes. To test this hypothesis, we examined the relationship between optimal codon usage in the classic galactose metabolism (*GAL*) pathway and known ecological niches for 329 species of budding yeasts, a diverse subphylum of fungi. We find that optimal codon usage in the *GAL* pathway is positively correlated with quantitative growth on galactose, suggesting that *GAL* codon optimization reflects increased capacity to grow on galactose. Optimal codon usage in the *GAL* pathway is also positively correlated with human-associated ecological niches in yeasts of the CUG-Ser1 clade and with dairy-associated ecological niches in the family Saccharomycetaceae. For example, optimal codon usage of *GAL* genes is greater than 85% of all genes in the genome of the major human pathogen *Candida albicans* (CUG-Ser1 clade) and greater than 75% of genes in the genome of the dairy yeast *Kluyveromyces lactis* (family Saccharomycetaceae). We further find a correlation between optimization in the *GAL*actose pathway genes and several genes associated with nutrient sensing and metabolism. This work suggests that codon optimization harbors information about the metabolic ecology of microbial eukaryotes. This information may be particularly useful for studying fungal dark matter—species that have yet to be cultured in the lab or have only been identified by genomic material.

(https://doi.org/10.6084/m9.figshare.c.4498292). Additional sequence analyses generated in this project, including the reference and annotated gene sequences, are stored in the figshare repository associated with this manuscript (https://doi.org/10.6084/m9.figshare.c.5067962). Data associated with each figure can be found in Supplementary Information file 4. All other information and data generated are available in the supplementary files.

**Funding:** This work was supported by the National Science Foundation under Grant Nos. DEB-1442113 (to A.R.) and DEB-1442148 (to C.T.H.), in part by the DOE Great Lakes Bioenergy Research Center (DOE BER Office of Science DE-SC0018409), USDA National Institute of Food and Agriculture (Hatch Project 1020204 to C.T.H.), and a Guggenheim fellowship (to A.R). C.T.H. is a Pew Scholar in the Biomedical Sciences and a H. I. Romnes Faculty Fellow, respectively supported by the Pew Charitable Trusts and the Office of the Vice Chancellor for Research and Graduate Education with funding from the Wisconsin Alumni Research Foundation. The funders had no role in study design, data collection and analysis, decision to publish, or preparation of the manuscript.

**Competing interests:** The authors have declared that no competing interests exist.

**Abbreviations:** *GAL*, galactose; GPI, glycosylphosphatidylinositol; KEGG, Kyoto Encyclopedia of Genes and Genomes; KO, KEGG Orthology; PGLS, phylogenetic generalized least squares; PIC, phylogenetically independent contrasts; stAI, species-specific tRNA adaptation index; YPD, yeast extract peptone dextrose.

## Introduction

The immense diversity of life is due, in part, to adaptation to the wide variety of environmental niches available. By acting on the interface between genotype, phenotype, and environment, natural selection has given rise to numerous ecological adaptations [1–3]. Elucidating the precise relationship between genotype, phenotype, and environment, however, is often challenging. For example, a connection was only recently made between environmental distribution of seeds of different sizes, phenotypic variation in the beaks of Darwin's finches, and changes in the expression of the protein BMP4 [4–6].

Genomic sequencing has accelerated the rate at which the underlying genomic mechanisms of well-established ecologically adapted phenotypes are elucidated [7,8]. While powerful, this type of ecological genomics requires extensive knowledge of the ecological niche in which species live. For many microbial species, however, detailed ecological information is unavailable due to both the scale of the ecosystems they live in and the dearth of information reported during collection [9]. One potentially powerful way to address this gap in knowledge is to use the extensive genomic resources available in microbes to conduct reverse ecology—directly predicting ecology from genotype [10,11].

Reverse ecology has been successful in broadly linking environmental phenotype with genotype using multiple types of genomic features [11–13]. Optimal growth temperature was successfully inferred from genomic content, including tRNA, ribosome, and gene features, in 549 Bacteria and 170 Archaea [14]. In the red bread mold *Neurospora crassa*, analysis of highly divergent genomic regions in 48 isolates uncovered "genomic islands" associated with adaptation in 2 different ecosystems [15]. Across the entire tree of life, metabolic capability (assessed using Kyoto Encyclopedia of Genes and Genomes (KEGG) gene annotations) was used to examine the evolution of exogenously required metabolites likely found in the environment [16]. Metabolic network analysis has emerged as a common genomic feature for reverse ecology analysis [17,18]. There are, however, other promising genomic features that can be used in reverse ecology.

One potentially useful but underutilized genomic feature for reverse ecology studies is codon usage, which has long been associated with gene expression [19–21]. Changes in gene expression have been shown to play an important role in ecological adaptation [22–24]. For example, in wild isolates of budding yeast *Saccharomyces cerevisiae*, changes in the expression of multiple genes were linked to phenotypic differences in copper resistance and pigmentation, which in turn may be associated with adaptation to high copper environments [25]. Over evolutionary time, increased levels of gene expression result in a selective pressure for accurate and efficient translation [26–30] and increased mRNA stability [31,32]. Codons that match the tRNA pool—called optimal codons—have a substantial impact on both translation [27,29,30] and mRNA stability [31]. Therefore, optimal codon usage is correlated with high gene expression in multiple lineages, especially in microbes [19,33–38]. Moreover, codon usage has both a mechanistic role (as a contributing factor to gene expression) and an evolutionary role (as an adaptation of moderately or highly expressed genes over time) within organisms. Therefore, we hypothesize that ecological adaptations that are, at least partly, due to high expression levels of specific genes or pathways will be reflected in their codon usage values.

Previous work in diverse microbes supports the hypothesis that codon optimization can be used to identify associations between codon usage and ecology [12,39–43]. For example, an analysis of metagenomes collected from mine biofilms shows an enrichment of optimal codons in bacterial and archaeal genes associated with inorganic ion transport [39]. In fungi, codon optimization in host-induced and secreted proteins is associated with generalist fungal parasites [41]. Although these studies were highly successful in linking particular ecological

niches with highly enriched groups of genes, we still lack examples where reverse ecology has linked particular ecologies to specific metabolic pathways.

The galactose (*GAL*) pathway (also known as the Leloir pathway) in the budding yeast subphylum Saccharomycotina is an iconic pathway that metabolizes galactose into glucose-1-phosphate, which can then be used in core metabolism or as an intermediate in other metabolic processes [44,45]. The genes encoding the 3 enzymes of the *GAL* pathway—*GAL1* (encoding a galactokinase), *GAL10* (encoding a UDP-glucose-4-epimerase), and *GAL7* (encoding a galactose-1-phosphate uridyl transferase)—are frequently clustered in yeast genomes and are induced in response to the presence of galactose [46–48]. There has been extensive research into the biochemistry [44], regulation [49–51], and evolutionary history [48,52] of this pathway. Ecological work on the *GAL* pathway revealed that gene inactivation is associated with an ecological shift in *Saccharomyces kudriavzevii*, a close relative of the species to *S. cerevisiae* [53]. There is also a positive association between galactose metabolism ability and the flower/*Ipomoea* isolation environment and a negative association between galactose metabolism ability and tree or insect frass isolation environments [54]. By identifying organisms that can grow in particular substrates, analyses of gene gain and loss give us insights into ecological adaptation; however, such analyses do not tell us how well organisms grow in these substrates—addressing this question requires understanding of variation in gene expression of the genes involved [55–57]. The recent publication of 332 budding yeast genomes and the identification of translational selection on codon usage in a majority of these species provide a unique opportunity to test for differences in *GAL* gene expression—inferred from optimal codon usage—across ecological niches inferred from recorded isolation environments [54,58–60].

In this study, we characterize the presence and codon optimization of the *GAL* pathway in 329 budding yeast species and identify an association between optimization in the *GAL* pathway and 2 specific ecological niches. We identify a complete set of *GAL* genes in 210 species and evidence of physical clustering of *GAL1*, *GAL7*, and *GAL10* in 150 species. Consistent with our hypothesis that codon optimization is a signature of high gene expression, we find that growth rate on galactose-containing medium is positively and significantly correlated with *GAL* codon optimization. In the CUG-Ser1 major clade, which contains the opportunistic human pathogen *Candida albicans*, codon optimization in the *GAL* pathway is higher in species found in human-associated ecological niches when compared to species associated with insect (and not human) ecological niches. In the family Saccharomycetaceae, another major clade in the subphylum Saccharomycotina that contains the model species *S. cerevisiae*, we find that codon optimization in the *GAL* pathway is higher in species isolated from dairy-associated niches compared to those from alcohol-associated niches. For example, codon optimization among closely related *Kluyveromyces* species is nearly twice as high in species isolated from dairy niches as those found associated with marine or fly niches. We also used KEGG Orthology (KO) annotations to find genes whose codon optimization correlated with *GAL* optimization. Many of the top hits in this analysis are known to be involved in sensing or regulating sugar metabolism in yeasts. These results suggest that codon optimization is a valuable genomic feature for linking metabolic pathways with ecological niches in microbes.

## Methods

### Galactose (*GAL*) pathway characterization

Genomic sequence and gene annotation data were obtained from the comparative analysis of 332 budding yeast genomes [58] (S1 Data). Mitochondrial sequences were filtered from these genomes using previously described methods [59]. Reference protein sequences for *GAL* gene

annotation (approximately 40 proteins for each of the *GAL* genes) and for *PMI40* gene annotation, which is involved in mannose assimilation and was used as a negative control (315 reference proteins), were obtained from GenBank and previous KO annotations [58,61]. A protein HMM profile was constructed for each gene and used to conduct 2 HMMER searches (version 3.1b2; http://hmmer.org/), one on publicly available annotations and one on all possible open reading frames generated using ORFfinder (version 0.4.3; https://www.ncbi.nlm.nih.gov/orffinder/). The search on all possible open reading frames was done to ensure that inferences of gene absences were not due to errors in publicly available gene annotations. The results of the 2 searches were compared using the Perl script fasta_uniqueseqs.pl (version 1.0; https://www.ncbi.nlm.nih.gov/CBBresearch/Spouge/html_ncbi/html/fasta/uniqueseq.cgi). Discrepancies between the 2 searches, which most often occurred in cases where the publicly available annotation combined 2 nearby genes, were resolved manually. The genes *GAL1* and *GAL3* are known ohnologs (i.e., paralogs that arose from a whole genome duplication event) [62,63]. We annotated ohnologs as *GAL3* only in the lineage where previous work has demonstrated a divergent function [51]. All other *GAL1* homologs were included in subsequent analyses as there is a lack of evidence for functional divergence [51]. All reference and annotated *GAL* and *PMI40* genes are available in the supplementary FigShare repository. All instances where *GAL1*, *GAL7*, and *GAL10* were found on the same contig were considered to represent *GAL* gene clusters.

## Codon optimization in the *GAL* pathway

To infer gene expression in the *GAL* pathway, we calculated the species-specific level of codon optimization in each *GAL* gene and compared it to the genome-wide distribution of codon optimization. Codon optimization of individual *GAL* genes was assessed by calculating the species-specific tRNA adaptation index (stAI) from previously calculated species-specific codon relative adaptiveness (wi) values [59,64]. Briefly, the species-specific codon relative adaptiveness values were calculated using stAI-calc [65] based on the genomic tRNA counts, optimized tRNA wobble weights, and genome-wide codon usage. Three species that previously failed to generate reliable wi values (*Martiniozyma abiesophila*, *Nadsonia fulvescens* var. *elongata*, and *Botryozyma nematodophila*) [59] were removed from all subsequent analyses. The stAI software does not take into account the CUG codon reassignment in the CUG-Ser1 and CUG-Ala clades. Previous analysis, however, suggests that this codon is rare [59]—the average frequency of the CUG codon in species where it has been reassigned is 0.005, 0.003, and 0.006 for *GAL1*, *GAL10*, and *GAL7*, respectively—and its influence on codon optimization calculations is not significant.

The stAI or species-specific tRNA adaptation index for each gene was calculated by taking the geometric mean of the previously calculated wi values for all the codons, except the start codon. The genome-wide distribution of gene stAI values is normally distributed, but the mean varies between species [59]. To compare codon optimization between species, we normalized each gene's stAI value using the empirical cumulative distribution function to get the percentage of all genes with stAI values lower than that of the gene of interest; we call this the estAI value. For example, an estAI value of 0.4 for a given gene would indicate that 40% of the genes in the genome have lower stAI values (i.e., are less optimized) than the gene of interest. The estAI optimization values therefore range from 0 to 1, with 1 being the most optimized gene in the genome. Codon optimization of the mannose metabolism gene *PMI40* was calculated in the same way as the *GAL* genes.

A total of 49 species' genomes had multiple copies of at least one *GAL* gene. For those genomes, the gene with the highest estAI value was used. For example, we identified 2 copies

of *GAL10* in *Candida ponderosae* located on different contigs with estAI values of 0.46 and 0.44. Therefore, we used the estAI value of 0.46 as the representative *GAL10* value for this species. The average difference between the maximum and minimum estAI for multiple copies of *GAL1*, *GAL7*, and *GAL10* are 0.0948, 0.0007, and 0.0125. There were 14 cases where all gene copies with the highest estAI values were not found on the same contig. In 18 cases, all duplicates with the highest estAI values were located on the same contig. The use of the *GAL* gene copy with the highest estAI is supported by evidence in *S. cerevisiae* that functionally derived gene duplicates have reduced codon optimization, which is likely linked to an evolutionary trajectory toward novel functions [66]. To test the assumption that the majority of ohnologs have diversified in function or are undergoing decay and are therefore unrelated to growth rate, we examined the relationship between copy number and growth rate. We conducted a comparative analysis using generalized estimating in R using the package ape (v5.3) [67,68], which allowed us to phylogenetically compare categorial data (single versus multiple *GAL* genes) and continuous data (growth rate). Within the Saccharomycetaceae, the presence of multiple *GAL* gene copies showed no significant correlation with growth rate on galactose containing medium (*p*-values of 0.85, 0.72, and 0.06 for *GAL1*, *GAL10*, and *GAL7*; full analysis on FigShare).

## Galactose growth data

To test the hypothesis that high levels of *GAL* codon optimization are associated with strong growth in media where galactose is the sole carbon source, we measured galactose, mannose, and glucose (as a control) growth for 258 species in the laboratory. Yeast strains corresponding to the species whose genomes were sequenced were obtained from the USDA Agricultural Research Service (ARS) NRRL Culture Collection in Peoria, Illinois, United States of America, or from the Fungal Biodiversity Centre (CBS) Collection in the Netherlands. All strains were initially plated from freezer stocks on yeast extract peptone dextrose (YPD) plates and grown for single colonies. YPD plates were stored at 4°C until the end of the experiment. To quantify growth on galactose, mannose, and glucose, we set up 3 replicates on separate weeks using different colonies for each strain. Strains were inoculated into liquid YPD and grown for 6 days at room temperature. For each replicate, strains were randomized and arrayed into a 96-well plate. The plate was then used to inoculate strains into a minimal medium containing 1% carbon source (D-galactose, 1% mannose, or 1% glucose), 5 *g*/L ammonium sulfate, and 1.7 *g*/L Yeast Nitrogen Base (w/o amino acids, ammonium sulfate, or carbon) and grown for 7 days. After a week, we transferred all strains to a second 96-well plate containing fresh minimal medium containing the carbon source.

To quantify the growth of each strain/species, we measured its optical density (OD units at 600 nm) following growth in a well of a BMG LABTECH FLUOstar Omega plate reader after a week at room temperature. We calculated 2 measures of growth, growth rate and endpoint, for each species and replicate. The growth rates were calculated in R (x64 3.5.2) using the grofit package (v 1.1.1.1), and end point, a proxy for saturation, was calculated by subtracting the $T_0$ time point from the final time point for each species. The growth rate was calculated as the maximum slope of the growth curve (S2 Fig) We visually assessed growth on galactose and mannose for all species using the growth curves we collected; a species was denoted as having the ability to grow on galactose or mannose if it grew in at least 2 of 3 replicates tested. Growth data, both growth rate and endpoint, were set to 0 for all species that did not meet this requirement. Quantitative growth on galactose was successfully measured for a total of 258 species. Growth on galactose or mannose was then computed relative to glucose, by dividing the growth rate by the rate measured on galactose, to account for differences in the baseline

growth rate of different species due to variables, such as cell size and budding type (unipolar versus bipolar). Therefore, a growth rate on galactose- (or mannose-) containing medium of 1 indicates that this species grew equally as quickly on galactose (or mannose) and glucose, whereas a growth rate of 0.5 indicates this species grew half as quickly on galactose (or mannose) as it did on glucose.

For the 71 species where new quantitative galactose growth data were unavailable, we used previously published species-specific binary growth data [54,58,60]. Uncertain growth is indicated where conflicting or variable growth was found in the literature (empty green triangles; Fig 1B). Quantitative galactose growth data (normalized to glucose) were compared to maximum gene codon optimization values using phylogenetically independent contrasts (PIC) [69]. Data from related species are not independent observations and therefore require a PIC analysis to ensure that covariation between traits is not the result of the relatedness of species [69]. The PIC analysis was conducted in R using the ape package (v5.3) [68]. The species *Metschnikowia matae* var. *matae* was removed from this analysis as it was a clear outlier on the residual plots for a complementary phylogenetic generalized least squares (PGLS) analysis (S3 Fig) [70,71]. Outliers in phylogenetically independent analyses occur when 2 closely related taxa have disparate trait values, which can be identified by examining the residual plots. In this case, the taxa *Metschnikowia matae* var. *maris* (strain yHMPu5000040795 = NRRL Y-63737 = CBS 13985) and *Metschnikowia matae* var. *matae* (strain yHMPu5000040940 = NRRL Y-63736 = CBS 13986) are very closely related, and yet the growth rate on galactose for *Metschnikowia matae* var. *matae* (1.390) is nearly double that of its closest relative *Metschnikowia matae* var. *maris* (0.750). The next most closely related species *Metschnikowia lockheadii* had a growth rate most similar to *M. matae* var. *maris* (0.567).

As negative controls of the specificity of the association between gene codon optimization and growth rate, we also compared the optimization of the mannose metabolism gene *PMI40* to growth rate on mannose- and galactose-containing media as well as the codon optimization of the *GAL* genes to growth rate on mannose.

## Prediction of growth rate from unsampled genomes

To test the predictive ability of the observed correlation between *GAL* codon optimization and growth rate on galactose-containing medium, we analyzed the publicly available genomes of 2 species not included in the original 332 species analyzed: *Kluyveromyces wickerhamii* (GCA_000179415 [73], UCD 54–210) and *Wickerhamiella occidentalis* (GCA_004125095, NRRL Y-27364 [74]). To obtain the stAI for each species, we first annotated each genome using AUGUSTUS [75] (v 3.2.3) trained on *S. cerevisiae* gene models. We then filtered out any coding sequences shorter than 500 nucleotides. The remaining coding sequences were then run through stAI-calc to obtain the species-specific wi values for each codon. Annotations and genes can be found on the FigShare repository.

We tested for translational selection on codon usage (S-test), annotated the *GAL* genes, calculated the estAI values for the *GAL* genes, and measured growth rate on galactose-containing medium standardized to growth rate on glucose-containing medium; we used the same approaches as for the analyses of the 332 genomes. We then used PGLS regression to predict growth rate on galactose-containing medium from the codon optimization values observed in each gene (we used the PGLS in this analysis because the PGLS transforms the regression, allowing us to estimate growth rate without transforming the raw growth data). These predictions can be improved in the future with the addition of a robust phylogeny including all species. We then compared how good our predictions were relative to the predictions made on simulated sequences whose codon assignment was random. To do this, we generated 1,000

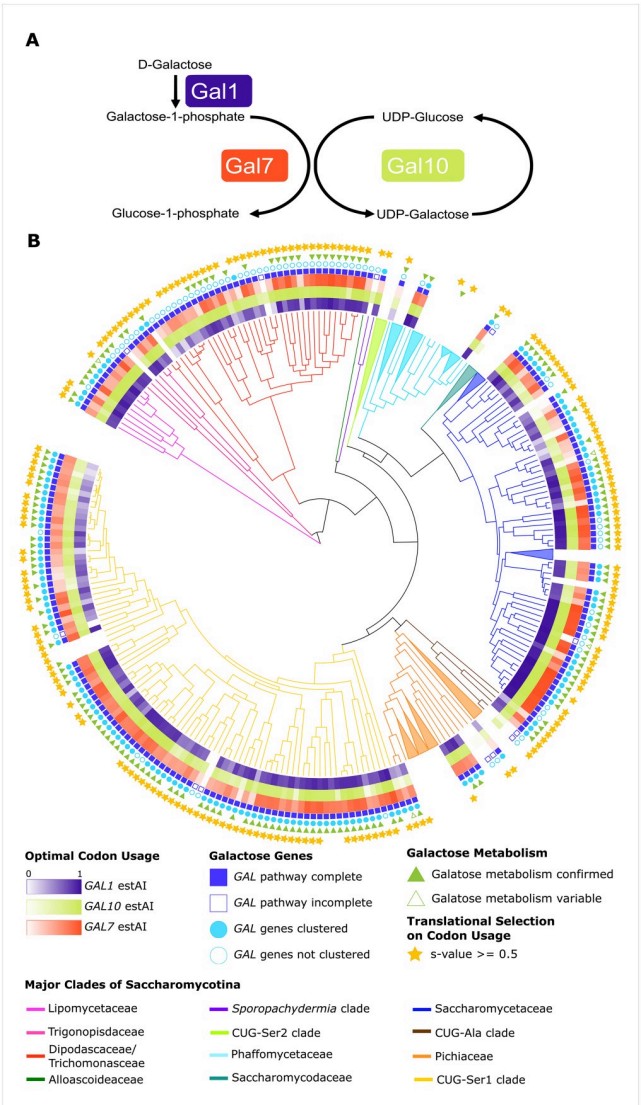

**Fig 1. The *GAL* pathway and the distribution of galactose metabolism, *GAL* genes, and preferred codon usage across the Saccharomycotina.** (A) The 3 enzymes of the *GAL* pathway metabolize galactose into glucose-1-phosphate, which can then enter glycolysis after being converted into glucose-6-phosphate. (B) Various features of galactose metabolism plotted on a phylogeny of the budding yeast subphylum Saccharomycotina; the 12 major clades of the subphylum are color coded. The presence and codon optimization (measured by estAI) of the 3 *GAL* genes are represented in the inner 3 rings. Clades with 3 or more species without a complete *GAL* pathway were condensed and are shown as triangles; for the full tree, see S1 Fig. The *GAL* clusters in the Dipodascaceae/Trichomonascaceae, Pichiaceae, and Phaffomycetaceae were recently found to have likely originated from horizontal gene transfer events from the CUG-Ser1 clade [72]. We did not identify any *GAL* genes from species in the CUG-Ser2 clade or the family Saccharomycodaceae. In every other major clade examined, we identified complete and clustered occurrences of the *GAL* pathway (filled-in blue squares and circles, respectively.) High codon optimization (darker colors) in the *GAL* pathway is not restricted to any one major clade. The ability to metabolize galactose (filled-in green triangle) was assessed either experimentally in this study or taken from the literature. In some instances, where only literature data were available, there were conflicting or variable reports of galactose metabolism (5 species; empty green triangles). The majority of species in the Saccharomycotina have also been shown to have genome-wide selection on codon usage (denoted by the yellow stars) [59]. Species names and ecological information can be found in S1 Fig and Table A S1 Data.

coding sequences with identical amino acid sequences to the observed *GAL* protein that had randomly assigned codon usage (data and custom perl script can be found on the FigShare repository). Based on the distribution of the growth rates of the random coding sequences, we then determined if the actual and predicted growth rate values fell within or outside the 95% confidence interval.

### Ecological association analysis

To test for associations between *GAL* pathway codon optimization and ecological niche, we obtained species-specific isolation data from multiple sources. We first tested 50 isolation environments from data collated from *The Yeasts*: *A Taxonomic Study* [60], as recorded by Opulente and colleagues [54,58]. We compared codon optimization in each of the *GAL* genes between species isolated from a given environment versus species not isolated from that environment (S4 Fig). From this analysis, we identified 4 general ecological niches with potentially differential codon optimization: dairy-, alcohol-, insect-, and human-associated ecological niches. To validate and update the data from *The Yeasts*, we conducted an in-depth literature search for these 4 specific ecological niches for each of the 329 species of interest using all known anamorphs and synonyms per species (see S2 Data for updated information for the ecological niches and associated references). Dairy ecological niches identified included milks, butters, cheeses, and yogurts. Alcohol ecological niches identified included spontaneous beer fermentation, alcohol starters, wine, ciders, kombuchas, and liquors. Insect-associated ecological niches included insect guts, insect bodies, and insect frass. Human-associated ecological niches were characterized as any isolation from a human, regardless of pathogenicity. We did not take into account studies where species identification lacked genetic data and relied solely on phenotypic and assimilation data, because these identifications have been shown to be potentially unreliable [76–78]. For example, the only evidence that the species *Candida castellii* is associated with dairy niches comes from a single identification in a fermented milk product using only metabolic characterization [79]. Therefore, *C. castellii* was not considered associated with dairy niches.

To test for significant differences in *GAL* optimization between ecological niches, we first filtered the species set to retain only those that contain all 3 *GAL* genes (210 species) and that were previously shown to exhibit genome-wide selection on codon usage (266 species; s-value $> = 0.5$) [59]; thus, the total number of species tested was 170. We then compared levels of *GAL* codon optimization between ecological niches using the Wilcoxon rank sum test in R [80].

### Evolutionary rate analysis

To examine variation in the evolutionary rates among both *GAL* and a representative set of genes in the genome, we used the maximum likelihood software PAML (version 4.9) [81,82]. Specifically, we examined the rates of synonymous changes in the *Kluyveromyces* species using the free-ratios model that allows for a different rate of evolution along each branch. The species tree was used as the input tree, and nucleotide sequences were aligned using the codon aware software TranslatorX (http://translatorx.co.uk/) [83]. For the analysis of a representative set of genes in the genome, we identified 651 previously annotated BUSCO genes [58] present in all four *Kluyveromyces* species for which the confidence interval was <10 for all PAML-inferred branch lengths. We then compared the distribution of BUSCO genes' branch lengths within each species with those observed for the *GAL* genes. These data can be found on the FigShare repository.

## Identification of additional metabolic pathways whose codon usage correlates with *GAL* optimization

To identify additional pathways that exhibit the same codon optimization trends between ecological niches as the *GAL* pathway, we tested whether the optimization of KOs was correlated with that of the *GAL* genes. KO annotations were previously generated for all species [58]. We started with the 266 genomes with evidence of translational selection on codon usage and identified 2,573 KOs present in 100 or more of those species. We then conducted a PIC analysis between the optimization of the *GAL* genes and each of the KOs across the species. *p*-values were adjusted to account for the total number of KOs tested using a Bonferroni correction (S3 Data).

# Results and discussion

## Variable *GAL* pathway and codon optimization across the Saccharomycotina

To examine variation in *GAL* codon optimization across the subphylum, we first examined whether *GAL* genes were present in each of the 329 genomes. Across the Saccharomycotina, we annotated 742 *GAL* genes (265, 256, and 221 annotations for *GAL1*, *GAL10*, and *GAL7*, respectively) in a total of 233 species (S1 Data and FigShare repository). The complete *GAL* enzymatic pathway (i.e., *GAL1*, *GAL10*, and *GAL7*) was identified in 210 species, of which 149 had evidence of *GAL* gene clustering. We cannot, however, rule out clustering of the *GAL* genes in the remaining 61 species as some of the annotations were at the ends of the contigs.

There were some discrepancies between galactose growth data and *GAL* gene presence data. Three species where galactose growth was experimentally observed lacked all 3 *GAL* genes: *Ogataea methanolica*, *Wickerhamomyces* sp. YB-2243, and *Candida heveicola*. The growth rates for these species are 0.129, 0.339, and 0.211 for *O. methanolica*, *Wickerhamomyces* sp., and *C. heveicola*. The low growth rates (seventh and third lowest overall) of *O. methanolica* and *C. heveicola* suggest these species may be utilizing trace amounts of other nutrients present in the medium. Finally, there were 26 species with a complete *GAL* gene cluster where no growth on galactose has been reported. This may represent a loss of pathway induction in these species or an inability to induce growth in the specific experimental conditions tested, as observed previously in the genus *Lachancea* [84]. Inactivation of the *GAL* pathway has also occurred multiple times in budding yeasts [48,53], and some of these taxa could be in the early stages of pathway inactivation.

Codon optimization in the *GAL* pathway, measured by estAI, varied greatly across the Saccharomycotina (Fig 1B.) The estAI values ranged from 0.02 (or greater than only 2% of the genes in the genome) in *GAL7* from *Lachancea fantastica* nom. nud. to 0.99 (or greater than 99% of the genes in the genome) in *GAL1* from *Kazachstania bromeliacearum*. To determine if there was an association between codon optimization and the ability to grow on galactose, we compared optimization in the *GAL* pathway between species that are able and unable to grow on galactose. We found that species without evidence for growth on galactose had significantly lower ($p < 0.05$) codon optimization in their *GAL1* and *GAL7* genes and that optimization in *GAL7* and *GAL10* was significantly lower ($p < 0.05$) in species with only a partial pathway (S5 Fig). Interestingly, the dataset contained no species with a partial *GAL* pathway containing *GAL1*. These correlations are consistent with a relaxation of selective pressures in nonfunctional pathways [85–87], and previous work has identified multiple parallel inactivation events of the *GAL* pathway in budding yeasts [53]. The *GAL* pathway may have alternative roles in cell function that are not associated with growth on galactose and may have not experienced

the same selective pressures. For example, in *C. albicans*, *GAL10* has been shown to be involved in cell integrity [88]. Finally, the *GAL* pathway may have an alternative induction system in these species. For example, the fission yeast *Schizosaccharomyces pombe* (not a member of the Saccharomycotina) has a complete *GAL* cluster but is unable to grow on galactose. Mutants of *S. pombe*, however, have been isolated that constitutively express the *GAL* genes and can grow on galactose [89].

### *GAL* codon optimization is correlated with growth rate on galactose

Strong translational selection on codon usage is correlated with highly expressed genes in diverse organisms [34,35,37,90–94]. Therefore, we hypothesized that high levels of codon optimization in the *GAL* pathway reflect high levels of *GAL* gene expression and ultimately high growth rates on galactose. Previous work in the closely related budding yeasts *S. cerevisiae* and *Saccharomyces uvarum* suggests that the more robust growth of *S. uvarum* on galactose is associated with multiple underlying genetic differences that make the *GAL* network more active in this species [51,52,84,95]. The 2 species' differences in growth rate on galactose and in *GAL* network activity are reflected in their *GAL* gene codon optimization values. Even though the *GAL* genes of *S. cerevisiae* and *S. uvarum* have high amino acid sequence identity (88.6%, 87.4%, and 88.6% for *GAL1*, *GAL10*, and *GAL7*, respectively), all 3 genes are more highly optimized in *S. uvarum* compared to *S. cerevisiae* (0.78 versus 0.70 for *GAL1*, 0.66 versus 0.46 for *GAL10*, and 0.55 versus 0.33 for *GAL7*). These results suggest that, over evolutionary time, a higher growth rate associated with increased flux through this pathway has resulted in a higher optimization of the *GAL* genes in *S. uvarum*.

Across the budding yeasts, we measured growth rate on galactose relative to glucose. We found a significant positive correlation between growth rate on galactose-containing medium and codon optimization in the *GAL* pathway of species that can grow on galactose and whose genomes have experienced translational selection on codon usage (N species = 94, linear regression of PIC values; *p*-values of 0.005, 0.012, and $3.207e^{-9}$ for *GAL1*, *GAL10*, and *GAL7*, respectively; Fig 2). Codon optimization of *GAL7* showed the strongest correlation with growth rate (Fig 2C), which may reflect the gene's function; *GAL7* encodes for the enzyme that metabolizes galactose-1-phosphate, a toxic intermediate [96,97] whose accumulation has been shown to reduce growth rate in *S. cerevisiae* [97]. Furthermore, the correlation between *GAL7* optimization and growth rate on galactose remained strong when analyzed independently in both the Saccharomycetaceae (29 species) and in the CUG-Ser1 clade (47 species; S6 Fig), the 2 largest clades sampled. Furthermore, the *GAL7* correlation was higher than additive models that included more than one gene—the second-best model correlated codon optimization with *GAL7* and *GAL10* with growth rate on galactose (*p*-value of $1.82e^{-8}$). The *GAL1* and *GAL10* genes were both significantly positively associated with growth rate in galactose in the Saccharomycetaceae, but not in the CUG-Ser1 clade (S6 Fig). This contrast may reflect the different regulatory mechanisms involved in galactose assimilation in the 2 major clades—tight control via a regulatory switch in the Saccharomycetaceae versus leaky expression in CUG-Ser1 [49,50]. We also tested the correlation between growth rate on galactose-containing medium and the *PGM1* and *PGM2* genes that encode phosphoglucomutases, enzymes that operate downstream of the galactose metabolism pathway by converting glucose-1-phosphate (Fig 1) to glucose-6-phosphate, which can enter glycolysis. There was no correlation between codon optimization in either *PGM1* or *PGM2* and growth on galactose-containing medium (S7 Fig). This result is likely due to the fact that both genes are involved in multiple metabolic pathways in addition to galactose metabolism.

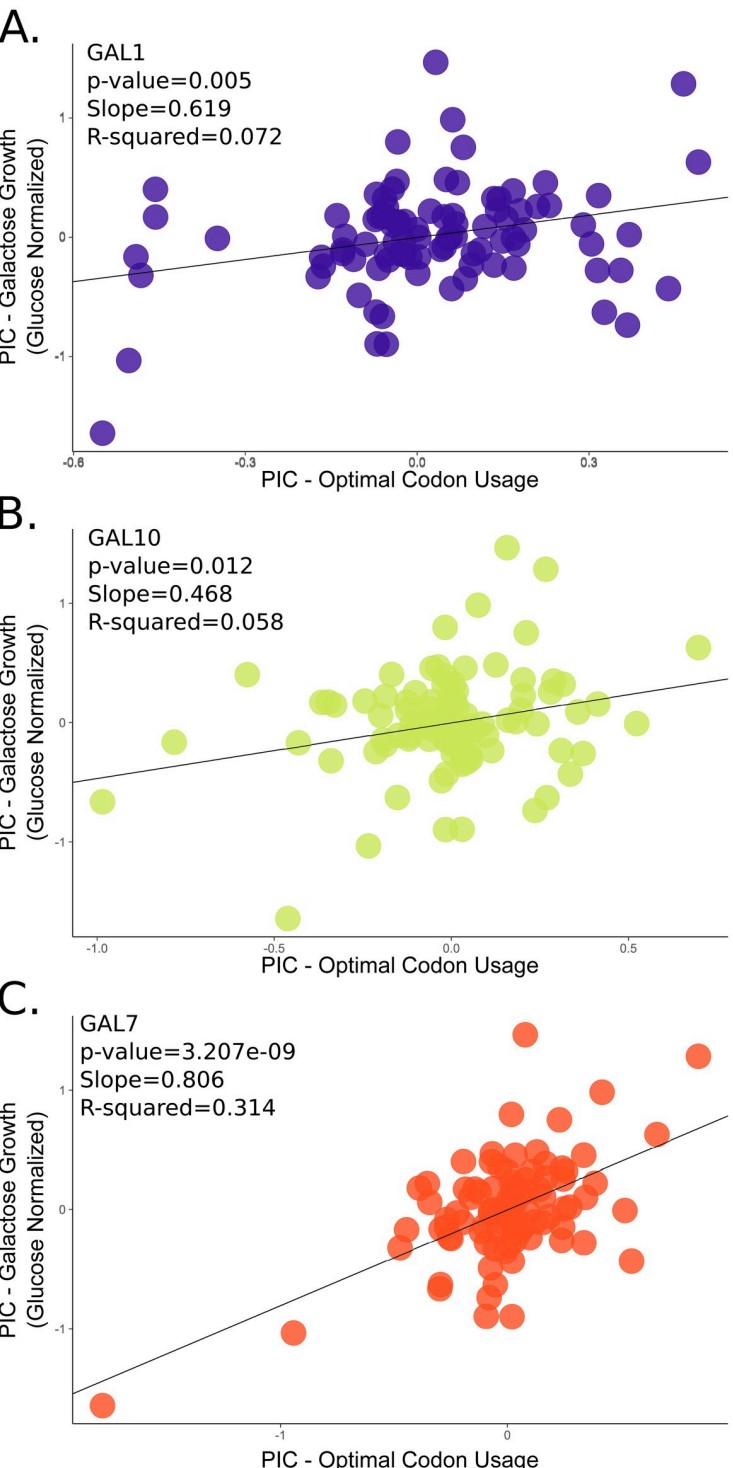

**Fig 2. Codon optimization in the *GAL* pathway is positively and significantly correlated with growth rate on galactose.** Phylogenetically independent contrasts (PIC) analyses of galactose growth (*y* axis) versus *GAL* gene optimal codon usage (*x* axis) in species that grow on galactose containing medium. Each point represents an independent contrast calculated between 2 species for either optimal codon usage or growth. A plot of the uncorrected data can be found in S11 Fig. There is a significant and positive correlation between the PIC values for codon optimization and galactose growth in *GAL1* (A), *GAL10* (B), and *GAL7* (C). The best fit and strongest correlation is between growth on galactose and optimization in *GAL7* (C). The analyses included 94 species with a growth rate on galactose greater than 0, a complete *GAL* cluster, and evidence of genome-wide translational selection on codon usage. One species,

*Metschnikowia matae* var. *matae*, was removed as an obvious outlier based on residual analysis. Underlying data can be found in Tables B–D S4 Data.

To test the specificity of the observed correlation between *GAL* genes and growth on galactose, we examined whether *GAL* genes correlated with growth on mannose-containing medium, as well as whether codon optimization of the well-characterized mannose metabolism gene *PMI40* correlated with growth on galactose [98]. Codon optimization of *PMI40* did not correlate with growth rate on galactose (*p*-value 0.61 and R-squared −0.0079; S8 Fig). Similarly, *GAL* gene optimization did not correlate with growth rate on mannose (*p*-values of 0.97, 0.92, and 0.11 for *GAL1*, *GAL10*, and *GAL7*, respectively; S9 Fig). Interestingly, the average *PMI40* codon optimization was very high (0.84) and not well correlated with growth rate on mannose (*p*-value 0.54 and R-squared −0.0048; S10 Fig). This suggests not only that *PMI40* is highly expressed regardless of metabolic state but also that optimization in some pathways may better correlate with growth rate than in others. Nevertheless, these results illustrate that the codon optimization in *GAL*actose metabolism genes is specifically correlated with growth rate on galactose. Collectively, our findings support the hypothesis that codon optimization is the result of selection on codon usage in species with high *GAL* gene expression.

## Can we predict growth rate on galactose from *GAL* codon optimization data?

To examine if our analysis enables the prediction of growth rate on galactose-containing media, we analyzed the codon optimization of the *GAL* pathway in 2 species not included in the original dataset—*K. wickerhamii* (Saccharomycetaceae) and *W. occidentalis* (Dipodascaceae/Trichomonascaceae clade). Both species have evidence of translational selection on codon usage based on the S-test (s-values of 0.58 for *K. wickerhamii* and 0.68 for *W. occidentalis*). The growth rate values predicted for *K. wickerhamii* from the *GAL1*, *GAL10*, and *GAL7* codon optimization values were 0.65, 0.44, and 0.47, respectively (S11 Fig). The growth rate of *K. wickerhamii* measured in the laboratory was 0.58. For *W. occidentalis*, we predicted growth rate values of 0.53, 0.54, and 0.76 based on *GAL1*, *GAL10, and GAL7* codon optimization values, respectively (S11 Fig). The growth rate measured in the laboratory was 0.86.

To better understand the performance of our growth rate predictions based on codon optimization values, we tested how well *GAL* genes with randomly assigned codons could predict growth rate. For both species, the predicted and the actual growth rates fell outside of the 95th percentile (S12 Fig). Additionally, for all but one observation (*GAL7* in *K. wickerhamii*), the predictions and actual growth rate fell outside the 99th percentile. These results suggest that the prediction of galactose growth rate from *GAL* codon optimization in budding yeasts is highly informative. Future sequencing and growth characterization currently underway across the Saccharomycotina will provide additional data for model development and validation.

## *GAL* codon optimization is associated with specific ecological niches

We further hypothesized that adaptation to specific ecological niches is associated with increased expression of the *GAL* pathway. To identify possible associations between *GAL* optimization and ecology, we conducted preliminary tests across 50 previously characterized ecological niches [54,60] for 114 species. This analysis led to an extensive literature search for 4 ecological niches of interest—dairy, alcohol, human, and insect—to maximize the number of species with ecological information. We uncovered 2 examples of niche-specific codon optimization (Fig 3): in the CUG-Ser1 clade, we found that *GAL* gene optimization was significantly

## A. CUG-Ser1 Clade

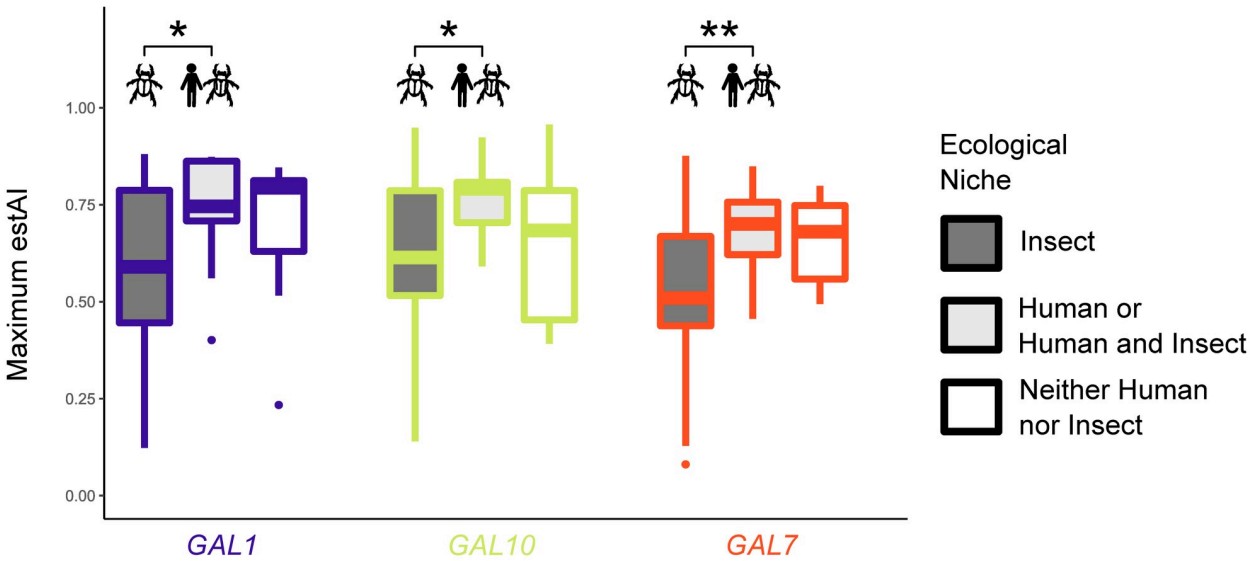

## B. Saccharomycetaceae

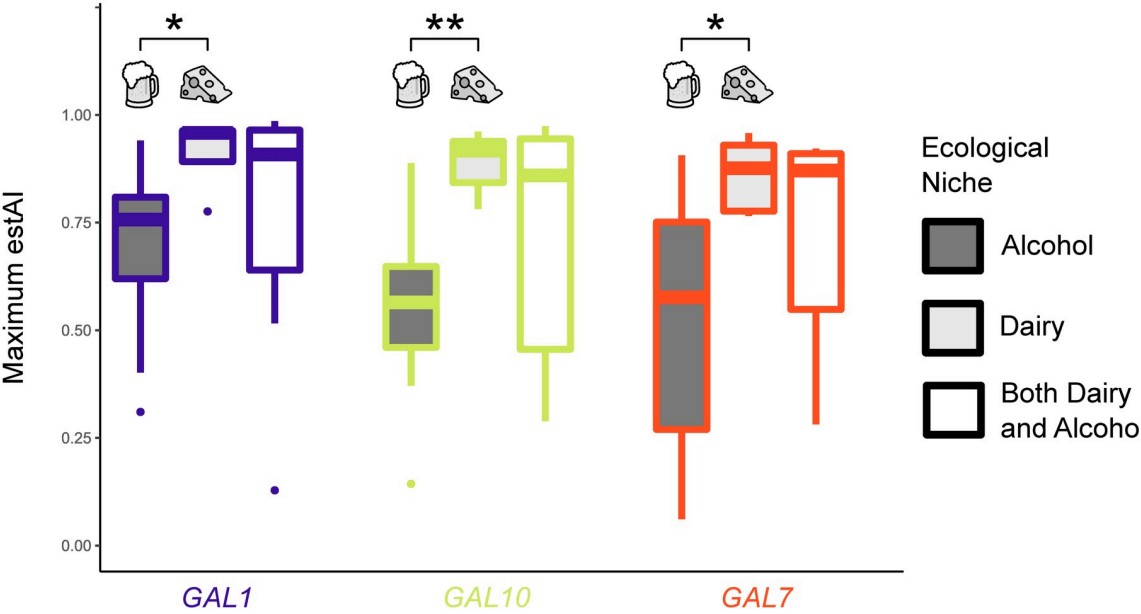

**Fig 3. Codon optimization in the *GAL* pathway is correlated with specific ecological niches in 2 different major clades of budding yeasts.** *p*-values less than 0.01 are indicated with ** and less than 0.05 with *. (A) In the CUG-Ser1 clade, species associated with a human niche or human and insect niches (13 species) have significantly higher codon usage optimization values in all *GAL* genes (*p*-values of 0.022, 0.028, and 0.006 for *GAL1*, *GAL10*, and *GAL7*, respectively) when compared to species that are associated with insect niches but not human niches (44 species). Only 11 species were not associated with either human or insect niches. (B) In the Saccharomycetaceae, species associated with only dairy niches (5 species) have significantly higher codon usage optimization values in all of the *GAL* genes (*p*-values of 0.010, 0.002, and 0.014 for *GAL1*, *GAL10*, and *GAL7*, respectively) versus species associated with only alcohol niches (14 species). A total of 9 species are associated with both dairy and alcohol niches. Underlying data can be found in Tables E and F S4 Data.

higher in species that have been isolated from human-associated ecological niches versus those that have been isolated from insect-associated niches; and in the Saccharomycetaceae, we found *GAL* gene optimization was significantly higher in species isolated only from dairy-associated niches compared to species isolated only from alcohol-associated niches. Differential codon optimization between environments could be due to selection for increased optimization, a relaxation of selection, or selection for nonoptimal codon usage. While we are unable to differentiate between these scenarios, the elevated codon optimization of the *GAL*actose genes relative to each species' genome-wide distribution suggests that translational selection on codon usage has occurred in this pathway.

*CUG-Ser1 clade*: Among CUG-Ser1 species that exhibit high genome-wide evidence of translational selection on codon usage (s-value ≥ 0.5), we found that *GAL* gene optimization was significantly higher ($p < 0.05$) in species from human-associated ecological niches or human- and insect-associated niches versus those that have been isolated from insect-associated niches only (57 species; Fig 3A). Only 2 species were found in human-associated niches and not insect-associated niches, *Debaryomyces subglobosus* and *Cephaloascus fragrans*; thus, we combined the human-associated species with the human- and insect-associated species into one group for subsequent analyses. Recent work has shown that many opportunistically pathogenic budding yeasts are likely to be associated with both environmental and human niches [99]. The 13 CUG-Ser1 species isolated from humans with genome-wide evidence of selection on codon usage had a mean optimization of 0.74, 0.76, and 0.69 for *GAL1*, *GAL10*, and *GAL7*, respectively.

We also found that *GAL1*, *GAL10*, and *GAL7* optimization was significantly higher (Wilcoxon rank sum test *p*-values of 0.035, 0.014, and 0.003, respectively) in species from human-associated ecological niches than insect-associated niches only, irrespective of genome-wide evidence of translational selection (88 species). For example, the major human pathogen *C. albicans* does not have genome-wide evidence for high levels of translational selection but has a very high *GAL10* codon optimization (estAI = 0.86). While *C. albicans* may not have evidence of genome-wide selection on codon optimization, a previous analysis suggests that at least 17% of genes in the *C. albicans* genome have likely experienced selection on codon usage [59].

Other opportunistic human pathogenic species with very high *GAL10* codon optimization (estAI > 0.8) include *Candida dubliniensis* [100], *Meyerozyma caribbica* [101], *Candida tropicalis* [102], *Meyerozyma guilliermondii* [103], and *Clavispora lusitaniae* [104]. The optimization of *GAL10* in human pathogenic species is consistent with findings that *GAL10* expression is up-regulated during *C. albicans* growth in the mammalian intestinal track [105]. Furthermore, *GAL10* in *C. albicans* is required for cell wall integrity, resistance to oxidative stress, and other virulence-related traits, even in the absence of galactose [88]. This suggests that *GAL10* may play an additional role, outside of galactose metabolism, in the CUG-Ser1 clade.

Interestingly, the highest *GAL10* optimization (average estAI = 0.93) in the CUG-Ser1 clade is found in *Spathaspora* species. While many *Spathaspora* species have been isolated from insects, 4 of the 5 species studied here (*Spathaspora girioi*, *Spathaspora hagerdaliae*, *Spathaspora gorwiae*, and *Spathaspora arborariae*) have been isolated only from rotting wood [106,107]. This observation is particularly interesting given the hypothesis that some features of saprophytic fungi, such as *Aspergillus fumigatus* and *Cryptococcus* spp., enable or predispose them to colonize human hosts [108,109]. Moreover, some pathogenic budding yeasts, including *C. albicans* and *C. tropicalis*, have recently been associated with soil [99]. Taken together, these results suggest that the *GAL*actose metabolism pathway may contribute to the capability of CUG-Ser1 species to opportunistically infect humans through metabolic shifts, cell wall

changes, or preadaptation to human-like environments. Further experimentation is needed to fully understand the contributions of this pathway to fungal pathogenicity.

*Saccharomycetaceae*: Among Saccharomycetaceae species, we found that *GAL* optimization is significantly higher ($p < 0.05$) in those that have been isolated only from dairy-associated niches compared to species isolated only from alcohol-associated niches (19 species; Fig 3B.) Only one species isolated from either dairy or alcohol, namely the alcohol-associated *Lachancea thermotolerans*, did not have evidence of genome-wide translational selection on codon usage. The 4 species isolated only from dairy-associated niches (*Kluyveromyces lactis*, *Naumovozyma dairenensis*, *Vanderwaltozyma polyspora*, and *Kazachstania turicensis*) have mean codon optimization values of 0.90, 0.88, and 0.84 for *GAL1*, *GAL10*, and *GAL7*, respectively. The 10 species that are only from alcohol-associated niches (S2 Data) have mean codon optimization values of 0.73, 0.61, and 0.59 for *GAL1*, *GAL10*, and *GAL7*, respectively. We also note that an association between galactose metabolism and these environments would not be apparent from gene presence/absence data only—a commonly used genomic feature in reverse ecology studies. Of the species found in dairy, alcohol, or both environments, the percentage of species with a complete *GAL* pathway was 83%, 88%, and 90% for the niches, respectively. The 4 dairy-associated species lacking a complete *GAL* pathway (the 17%) may be utilizing the other sugars and fermentation products available in dairy. Therefore, whereas gene presence/absence analyses may be better suited to address whether organisms "can" grow on a given substrate, codon optimization analyses may be better suited to address "how well" organisms grow on a given substrate. The association between *GAL* codon optimization and dairy associated niches may be the result of the fact that in many dairy environments, there are large microbial communities that often consist of lactic acid bacteria that convert lactose into glucose and galactose, which can subsequently be used in the *GAL* pathway [110,111]. The natural presence of galactose in dairy-associated environments is the likely driver of *GAL* codon optimization.

Species found in both dairy- and alcohol-associated niches have a range of optimization values that generally encompasses the values observed for species from dairy- or alcohol-only niches. It is likely that this group (associated with both dairy and alcohol niches) contains species or populations that are better adapted to one niche than the other. It is not possible, however, based on current literature to disentangle these 2 categories. For example, the species *Kluyveromyces marxianus* has been isolated from chica beer [112], cider [113], kombucha [114], and mezcal liquor [115]. However, *K. marxianus* is a well-known "dairy yeast" frequently found in both natural [116,117] and industrial dairy products [118]. Codon optimization of the *GAL* enzymatic pathway is also very high in *K. marxianus* with an average estAI of 0.92. We hypothesize that the high *GAL* codon optimization in *K. marxianus* is a result of its association with dairy and with the ability of *K. marxianus* to metabolize lactose into glucose and galactose [44]. There are 2 species that are associated with both dairy and alcohol niches whose *GAL* codon optimization values are higher than the maximum value observed in alcohol-only species—*Naumovozyma castellii* and *Kazachstania unispora*. Based on this, we hypothesize that these species are well adapted to dairy-associated environments.

The species *K. wickerhamii*, for which we predicted growth rate on galactose-containing media from codon optimization, belongs to the Saccharomycetaceae. The observed codon optimization values of *K. wickerhamii GAL* genes lie below the lower (first) quantile of species found in dairy-associated niches: *GAL1* below 0.892 (observed 0.887), *GAL10* below 0.843 (observed 0.660), and *GAL7* below 0.776 (observed 0.531). We would therefore predict that *K. wickerhamii* is not associated with dairy niches. Consistent with this prediction, *K. wickerhamii* is characterized as "confined to *Drosophila* species and to tree exudates that probably act as *Drosophila* habitats" [60].

## Differential *GAL* pathway optimization in *Kluyveromyces*

The genus *Kluyveromyces* provides an example of how codon optimization varies between closely related species that differ in their ecological niches (Fig 4). Two of the 4 species in this clade have not been isolated from either dairy or alcohol; *Kluyveromyces aestuarii* has been isolated from marine mud and seawater, while *Kluyveromyces dobzhanskii* has been isolated from flies, plants, and mushrooms [60]. Of the 4 species represented here, only *K. dobzhanskii* is not known to metabolize lactose into glucose and galactose [60]. While all 4 species are capable of growing on galactose, *GAL* gene codon optimization is much higher in the 2 species with dairy-associated ecological niches, *Kluyveromyces lactis* and *Kluyveromyces marxianus* (Fig 4A). Codon optimization for *GAL* genes is greater than 75% of the genome (estAI > 0.75) for *K. lactis* and *K. marxianus*. In *K. marxianus*, the optimization of *GAL1* and *GAL10* (estAI 0.93 and 0.94) is nearly that of the average ribosomal gene (estAI 0.99; Fig 4B). Ribosomal genes, which are among the most highly expressed genes in the genome, are known to be highly optimized in a broad range of species [119]. In contrast, optimization values for *GAL* genes in *K. aestuarii* and *K. dobzhanskii* are nearer to the mean (mean estAI values of 0.63 and 0.46, respectively; Fig 4B).

We hypothesized that the low *GAL* optimization in *K. aestuarii* and *K. dobzhanskii* was due to a relaxation in translational selection on the *GAL* pathway. To test this hypothesis, we estimated the rate of synonymous site evolution using PAML on the *GAL* genes and 651 BUSCO genes. In *K. aestuarii* and *K. marxianus*, the *GAL* genes all fell within the median 50% of BUSCO genes, suggesting that the synonymous sites in these species have not experienced rapid change or levels of conservation. In the sister species *K. dobzhanskii* and *K. lactis*, there was some evidence of relaxation of selection on synonymous sites in *GAL10* and *GAL1* in *K. dobzhanskii*, leading to longer branches. In *K. lactis*, *GAL7* also had a short branch that fell within the most slowly evolving 25% of the highly conserved BUSCO genes, which may indicate constraint on this gene's codon usage. This finding is consistent with the strong association between *GAL7* and growth rate on galactose-containing medium.

## *GAL* optimization is correlated with optimization in other metabolic genes

In general, multiple metabolic pathways, as opposed to a single one, likely contribute to adaptation to an ecological niche [54,120]. To identify genes in other metabolic pathways that are potentially associated with galactose metabolism, we tested whether levels of codon optimization in *GAL* genes were significantly correlated with levels of codon optimization in other KOs. We identified 78/2,572 KOs with a significant positive or negative association with *GAL* optimization (PIC, multiple test corrected *p*-value <0.05; S3 Data). To explore the results, we focused on the top hits for each individual *GAL* gene and hits that occurred in more than one *GAL* gene (Table 1).

Within the top correlated KO annotations, we found several annotations related to glucose and galactose metabolic sensing and regulation. It is important to note, however, that KEGG annotations are unable to differentiate between orthologs, ohnologs, or paralogs. Codon optimization of genes annotated with the KO K08139 were significantly and positively correlated with optimization in *GAL7*. Genes with this annotation have been implicated in metabolic sensing. This KO includes *SNF3* and *RTG2*, which have been shown to detect glucose in *S. cerevisiae* [121] but are associated with galactose sensing in *C. albicans* [122]. *GAL2*, the galactose transporter, is also associated with this annotation. To discern which homologs may have correlated expression with the *GAL*actose metabolism genes, further analysis and a more complete taxon sampling would be required. The annotation K05292 is associated with the *S. cerevisiae* gene *GPI16*, whose codon optimization was correlated with both *GAL7* and *GAL10*; *GPI16*

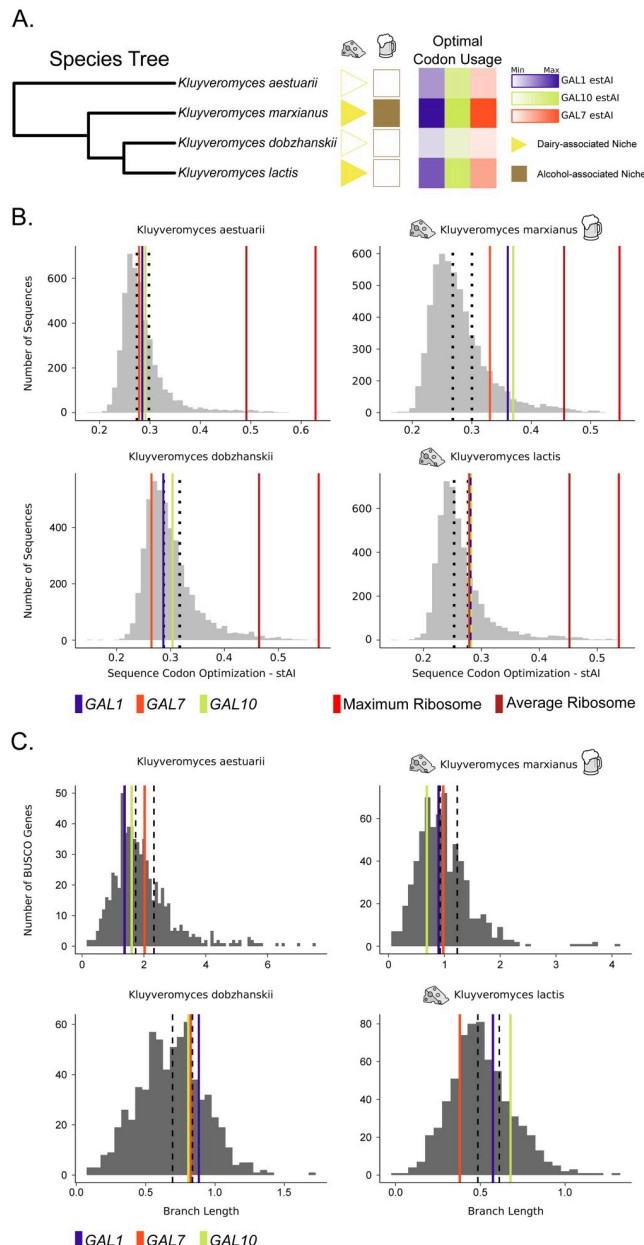

**Fig 4. Closely related *Kluyveromyces* species exhibit differential codon optimization in the *GAL* pathway associated with isolation from dairy environments.** All 4 *Kluyveromyces* species were shown experimentally to metabolize galactose. (A) Species phylogeny of 4 closely related *Kluyveromyces* species. *K. marxianus* and *K. lactis* are both associated with dairy niches and have high codon optimization values in their *GAL* pathway genes. In contrast, *K. aestuarii* is associated with marine mud, and *K. dobzhanskii* is associated with flies. (B) The genome-wide distribution of codon optimization (stAI) values for the 4 *Kluyveromyces* species included in this study. The 50th and 75th percentiles are shown with black dashed lines. In the 2 species associated with dairy niches, the codon optimization for all 3 *GAL* genes falls in the top 25th percentile. In the 2 species not associated with dairy, the *GAL* genes fall below the top 25th percentile. Genes encoding ribosomal proteins are well established to rank among the most highly optimized genes within a genome. (C) The distribution of terminal synonymous (dS) branch lengths (in unrooted gene trees) calculated for the 651 BUSCO genes. All 3 *GAL* genes fall within the interquartile range for *K. aestuarii* and *K. marxianus*. In *K. dobzhanskii*, all 3 *GAL* genes lie above the 70th percentile with *GAL1* in the upper quartile. In *K. lactis*, *GAL7* falls below the interquartile range, while *GAL10* lies above. The 50th and 75th percentiles are shown with black dashed lines. Underlying data can be found in Tables G–P S4 Data.

**Table 1. KEGG Orthology (KO) annotated genes that have a codon optimization correlated with *GAL1*, *GAL10*, or *GAL7*.** This table contains the significant (multiple test corrected *p*-value <0.001***) results with the greatest absolute slope for each *GAL* gene and 3 additional results that are significant in both *GAL7* and *GAL10*. Gene names for *S. cerevisiae* are listed where available.

| KO | KO Title | *S. cerevisiae* Gene | *GAL1* slope | | *GAL10* slope | | *GAL7* slope | |
|---|---|---|---|---|---|---|---|---|
| K00916 | *CTK1*; CTD kinase subunit alpha | *CTK1* | −0.28 | *** | −0.14 | | −0.17 | |
| K08139 | *HXT*; MFS transporter, SP family, sugar:H+ symporter | *RGT2, SNF3, GAL2, HXT1–11, HXT13–17* | 0.35 | | 0.42 | | 1.09 | *** |
| K02564 | *nagB*, GNPDA; glucosamine-6-phosphate deaminase | | 0.23 | | 0.65 | *** | 0.33 | |
| K05292 | PIGT; GPI-anchor transamidase subunit T | *GPI16* | 0.17 | | 0.52 | *** | 0.66 | *** |
| K14012 | NSFL1C, UBX1, SHP1; UBX domain-containing protein 1 | *SHP1* | 0.20 | | 0.49 | *** | 0.57 | *** |
| K17605 | PPP2R4, PTPA; serine/threonine-protein phosphatase 2A activator | *RRD1, RRD2* | −0.30 | | −0.47 | *** | −0.68 | *** |

encodes a subunit of the glycosylphosphatidylinositol transamidase complex. This complex is involved in the addition of glycosylphosphatidylinositol (GPI) anchors to proteins, which allow these proteins to be anchored to the plasma membrane [123]. Not only can these anchors contain galactose, but they may be involved in the recognition of opportunistic pathogens by the host immune system [123]. This correlation further supports the link between the galactose metabolism pathway and isolation from human associated niches in the CUG-Ser1 clade. Codon optimization in the gene *SHP1* is correlated with optimization of both *GAL7* and *GAL10*. Shp1 has been shown to be involved in glucose sensing in the cAMP-PKA pathway, which helps to regulate storage of carbohydrate levels [124]. Other notable genes whose optimization correlates with *GAL*actose genes are *YCK1/YCK2*, which are involved in glucose sensing [125]; *ICL1*, which encodes a member of the glyoxylate shunt that may be differentially used in alternative carbon metabolism [126], and *THI6/PHO8*, which is involved in the production of thiamine pyrophosphate, a cofactor used by several carbon metabolism enzymes. These results suggest that correlated codon optimization may be a useful way to computationally associate the expression of pathways. Follow-up analyses within specific subclades and experimental work are needed to determine whether these associations are mechanistically relevant.

## Conclusions

Here, we use reverse ecology to associate genotype (codon optimization) with phenotype (growth rate on galactose) and ecology (isolation environment) across an entire evolutionary lineage (budding yeasts). The conceptual evolutionary model for this association (Fig 5) is that selection for increased rates of galactose metabolism in galactose-rich environments will result in selection for optimization of codon usage in the *GAL* genes. This selection is likely to continue until codon usage is no longer a barrier to maximum flux allowed through this pathway for a given metabolic load. Therefore, codon optimization not only reflects a mechanistic measure of expression but an evolutionary signal for selection on increased expression.

By studying a well-known metabolic pathway in a diverse microbial subphylum, we provide a proof of concept for the utility of codon optimization as a genomic feature for reverse ecology. Our discovery of optimization in the *GAL* pathway in dairy-associated Saccharomycetaceae and human-associated CUG-Ser1 yeasts is consistent with the known functional roles of the enzymes in the pathway. The complete *GAL* pathway metabolizes galactose, a component of dairy environments, into usable energy [127]. The *GAL10* gene affects phenotypes associated with human colonization in CUG-Ser1 yeasts [88]. Similarly, in the *Kluyveromyces* species found on dairy-associated niches that are able to metabolize lactose into glucose and galactose,

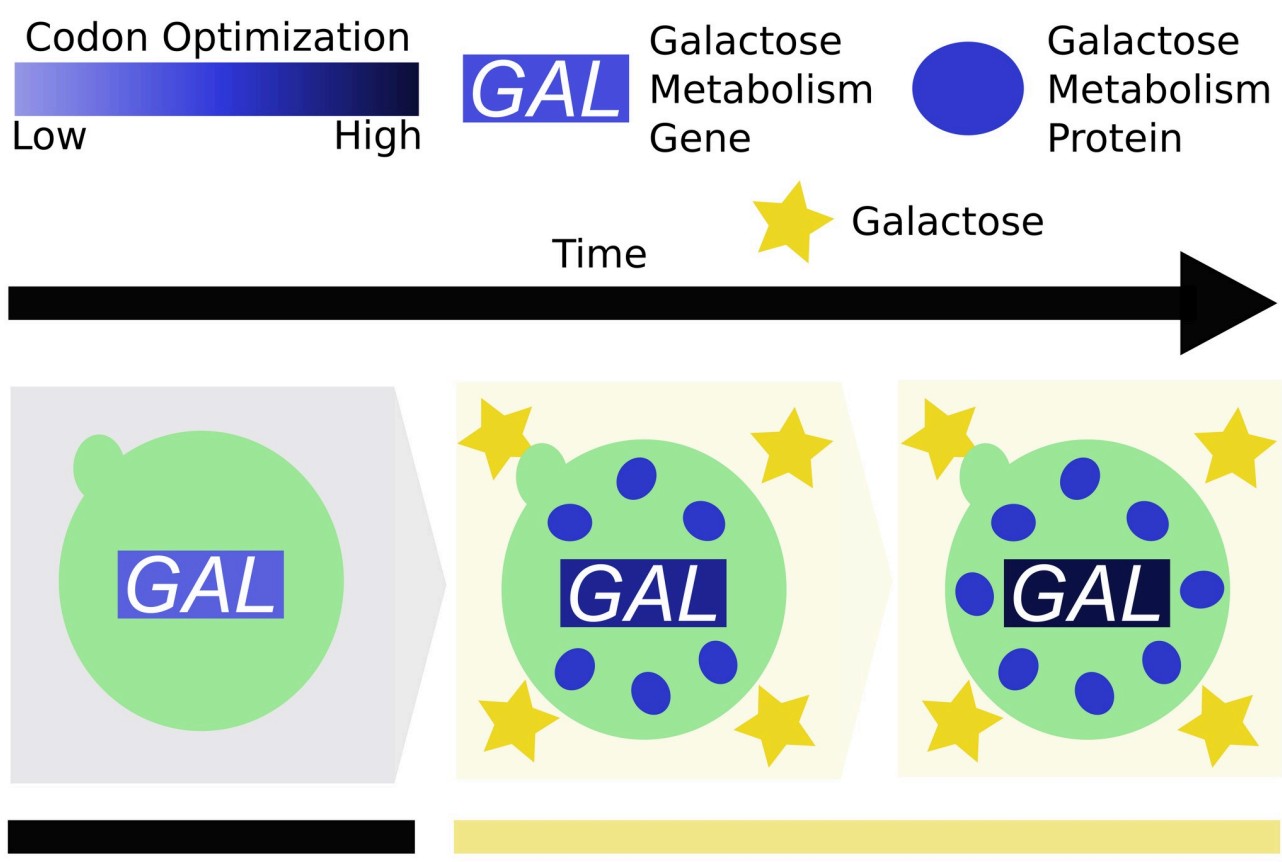

**Fig 5. A model for ecological codon adaptation in the *GAL*actose metabolism pathway.** The ancestral species in environment A maintains the *GAL*actose metabolism pathway at an intermediate codon optimization. Upon introduction to environment B, which contains abundant galactose, there is increased demand for the *GAL*actose metabolism enzymes to take advantage of this energy source. In this new environment, substitutions that increase codon optimization of the *GAL* genes will be selectively advantageous. Codon optimization will continue to increase under translational selection until it is no longer a barrier to expression or optimal flux through the pathway has been achieved.

there is high optimization in this pathway compared to closely related species not associated with dairy. Interestingly, examination of codon optimization in the gene sets of the 4 *Kluyveromyces* species studied here would have identified at least *K. marxianus* as a potential dairy-associated yeast, even in the absence of any knowledge about its isolation environments. Thus, genome-wide examination of codon optimization in fungal, and more generally microbial, species can generate specific hypotheses about metabolic ecology that can be experimentally tested. Our method can also be applied directly to single-cell genomes generated from microbial dark matter known only from DNA [128]. Finally, using an unbiased approach, we identified a strong correlation between optimization in the *GAL* pathway and other pathways involved in metabolic processing. This novel finding suggests that codon optimization may also be useful for identifying coregulated or correlated pathways in microbial, including fungal, species.

By focusing on a well-characterized pathway, we are able to associate a specific genotype with both a phenotype and ecology. While previous codon-based reverse ecology studies have identified functional categories of genes associated with environments [39–43], we illustrate that this approach can also be useful at the level of individual genes and pathways. It is

important to note that this approach may not work for all genes—especially those that have universally high codon usage, such as ribosomal genes and the mannose metabolism gene *PMI40*. More broadly, our results suggest that codon optimization can be a useful tool for predicting candidate genes and pathways involved in ecological adaptation, which can be subsequently tested experimentally.

## Supporting information

**S1 Fig. The *GAL* pathway and the distribution of galactose metabolism, *GAL* genes, and preferred codon usage across the Saccharomycotina.** Various features of galactose metabolism plotted on a phylogeny of the budding yeast subphylum Saccharomycotina; the 12 major clades of the subphylum are color coded. The presence and codon optimization (measured by estAI) of the 3 *GAL* genes are represented in the inner 3 rings. The *GAL* clusters in the Dipodascaceae/Trichomonascaceae, Pichiaceae, and Phaffomycetaceae were recently identified as likely originating from horizontal gene transfer events from the CUG-Ser1 clade. High codon optimization (darker colors) in the *GAL* pathway is not restricted to any one major clade. Complete and clustered occurrences of the *GAL* pathway are represented by filled-in blue squares and circles, respectively. Ecological associations were uncovered using a literature search (S2 Data).
(PDF)

**S2 Fig. Growth curve example for the species *Debaryomyces nepalensis*.** Growth rate is calculated based on the maximum slope of the curve in the R package grofit. The slopes calculated in this species for galactose are 0.0495, 0.0747, and 0.0862 in the replicates 1 to 3 for an average growth rate of 0.070. The slopes calculated in this species for glucose are 0.0712, 0.0762, and 0.0682 in the replicates 1 to 3 for an average growth rate of 0.072. Therefore, the glucose normalized rate of growth on galactose is 0.97.
(PDF)

**S3 Fig. Residual and Q-Q normalized plots of phylogenetic generalized least squares (PGLS) analysis of codon optimization and normalized growth rate on galactose containing medium.** (A) The residual versus fitted analysis shows 2 outlier strains: *Metschnikowia matae* var. *matae* and *Metschnikowia matae* var. *maris*. (B) Residual and Q-Q normalized plots after removal of *Metschnikowia matae* var. *matae*. No clear outliers remained after removal of this species.
(PDF)

**S4 Fig. Initial analysis of *GAL* codon optimization in species isolated from particular ecological niches versus those that have not been isolated from a niche.** Blue bars are the codon optimization values for species that have not been isolated from the particular ecology. Yellow bars are the codon optimization values for species that have been isolated from that ecology. Ecological information was tested in 50 isolation environments from data collated from *The Yeasts*: *A Taxonomic Study* as recorded by Opulente and colleagues [54].
(PDF)

**S5 Fig. Comparisons of optimal codon usage in *GAL* genes.** (A) Wilcoxon rank sum test of *GAL* codon optimization versus binary data for growth on galactose. A total of 170 species were included in this analysis. (B) Wilcoxon rank sum test of *GAL* codon optimization in species with complete or incomplete *GAL* pathways. A total of 185 species were included in this analysis.
(PDF)

**S6 Fig. PIC analysis of *GAL* codon optimization and growth rate on galactose containing medium (normalized to growth rate on glucose containing media).** (A) The phylogenetically corrected correlation between *GAL* codon optimization and quantitative growth on galactose-containing medium is significant in all genes when only species from the family Saccharomycetaceae are considered. A total of 29 species were included in this analysis. (B) The phylogenetically corrected correlation between *GAL* codon optimization and quantitative growth on galactose-containing medium is only significant in *GAL7* when only the CUG-Ser1 major clade species are considered. This analysis includes 47 species.
(PDF)

**S7 Fig. PIC analysis of correlation between growth rate on galactose containing medium and codon optimization in the gene *PGM1/PGM2*.** This analysis suggests that codon optimization in *PGM1/PGM2* does not contribute to growth on galactose-containing medium.
(PDF)

**S8 Fig. Codon optimization in the mannose metabolism gene PMI40 is not associated with growth rate on galactose-containing medium.**
(PDF)

**S9 Fig. Codon optimization in the *GAL*actose metabolism genes is not correlated with growth rate on mannose-containing medium.** This result supports the conclusion that the association between GAL optimization and growth rate on galactose is specific to that pathway.
(PDF)

**S10 Fig. Codon optimization in the mannose metabolism gene *PMI40* is not correlated with growth rate on mannose-containing medium.** This result is likely due to the very high codon optimization observed in *PMI40*.
(PDF)

**S11 Fig. Using PGLS linear correlations between codon optimization in the *GAL* genes and growth rate on galactose, we predicted the growth rate for 2 previously unanalyzed species: *Kluyveromyces wickerhamii* (blue data) and *Wickerhamiella occidentalis* (red data).** The circular points represent the predicted growth rates based on the observed codon optimization values (lines). The triangles represent the actual growth rate measured in the laboratory.
(PDF)

**S12 Fig. Predictions of growth rate on galactose medium from random codon *GAL* genes do not accurately predict growth rate measured in the laboratory.** To better understand the performance of our growth rate predictions based on codon usage, we tested how well *GAL*actose genes with randomly assigned codons could predict growth rate. The purple lines represent the empirically measured growth rate for each species. The orange line is the growth rate predicted by the actual codon usage for each gene based on the PGLS analysis. For each *GAL* gene associated with *K. wickerhamii* (panel A) and *W. occidentalis* (panel B), we generated 1,000 DNA sequences with random codon usage (while keeping the protein sequence identical). For each random sample, we then predicted growth rate based on our regression analysis. In both cases, both the predicted and the actual growth rates fall outside of the 95th percentile. Additionally, for all but one observation (*GAL7* in *K. wickerhamii*), the predictions and actual growth rate fall outside the 99th percentile. These results show that our predictions are highly informative.
(PDF)

**S1 Data. Species information for the studied Saccharomycotina.** This includes major clade, file name, *GAL*actose information, and codon selection values (S-value). (XLSX)

**S2 Data. Associated ecologies for the species studies.** This includes presence/absence data in addition to the corresponding references. (XLSX)

**S3 Data. Correlation of KEGG orthologous genes with *GAL*actose optimization.** For each KO, the correlation with each *GAL* gene is reported along with the multiple test corrected significance. Corrected *p*-values corresponding to 0.05, 0.005, and 0.001 are noted as *, **, and ***, respectively. (XLSX)

**S4 Data. Raw underlying data used to construct figures.** (A) Data for Fig 1, including species and *GAL*actose gene codon optimization. (B) Data for Fig 2A including *GAL1* codon optimization and growth rate on galactose relative to glucose per species. (C) Data for Fig 2B including *GAL10* codon optimization and growth rate on galactose relative to glucose per species. (D) Data for Fig 2C including *GAL7* codon optimization and growth rate on galactose relative to glucose per species. (E) Data for Fig 3A including each species, their *GAL* codon optimization and their ecological niche. This also includes the minimum, Q1, median, Q3, and maximum codon optimization values. (F) Data for Fig 3B including each species, their *GAL* codon optimization and their ecological niche. This also includes the minimum, Q1, median, Q3, and maximum codon optimization values. (G) Data for Fig 4A including the *Kluyveromyces* species, their codon optimization values and ecological niches. (H) Data for Fig 4b including *K. aesturaii* codon optimization values for all genes. (I) Data for Fig 4B including *K. aesturaii* codon optimization values for ribosomal genes. (J) Data for Fig 4B including *K. marxianus* codon optimization values for all genes. (K) Data for Fig 4B including *K. marxianus* codon optimization values for ribosomal genes. (L) Data for Fig 4B including *K. dobzhanskii* codon optimization values for all genes. (M) Data for Fig 4B including *K. dobzhanskii* codon optimization values for ribosomal genes. (N) Data for Fig 4B including *K. lactis* codon optimization values for all genes. (O) Data for Fig 4B including *K. lactis* codon optimization values for ribosomal genes. (P) Data for Fig 4C including the branch lengths modeled for each BUSCO gene analyzed. (XLSX)

## Acknowledgments

We thank the members of the Rokas and Hittinger labs for helpful discussions.

## Author Contributions

**Conceptualization:** Abigail Leavitt LaBella, Antonis Rokas.

**Data curation:** Abigail Leavitt LaBella.

**Formal analysis:** Abigail Leavitt LaBella, Dana A. Opulente.

**Funding acquisition:** Chris Todd Hittinger, Antonis Rokas.

**Investigation:** Abigail Leavitt LaBella, Dana A. Opulente, Jacob L. Steenwyk, Chris Todd Hittinger, Antonis Rokas.

**Methodology:** Abigail Leavitt LaBella.

**Project administration:** Chris Todd Hittinger, Antonis Rokas.

**Resources:** Dana A. Opulente, Chris Todd Hittinger, Antonis Rokas.

**Supervision:** Chris Todd Hittinger, Antonis Rokas.

**Validation:** Abigail Leavitt LaBella.

**Visualization:** Abigail Leavitt LaBella.

**Writing – original draft:** Abigail Leavitt LaBella, Antonis Rokas.

**Writing – review & editing:** Abigail Leavitt LaBella, Dana A. Opulente, Jacob L. Steenwyk,
Chris Todd Hittinger, Antonis Rokas.

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
