## [Editor Report · Decision Letter 0]

31 Jul 2020

Dear Dr Rokas, 

Thank you for submitting your manuscript entitled "Signatures of optimal codon usage predict metabolic ecology in budding yeasts" for consideration as a Research Article by PLOS Biology.

Your manuscript has now been evaluated by the PLOS Biology editorial staff, as well as by an Academic Editor with relevant expertise, and I am writing to let you know that we would like to send your submission out for external peer review.

Please re-submit your manuscript within two working days, i.e. by Aug 04 2020 11:59PM.

Kind regards,

Gabriel Gasque, Ph.D.,

Senior Editor

PLOS Biology

---

## [Decision Letter · Decision Letter 1]

1 Oct 2020

Dear Dr Rokas,

Thank you very much for submitting your manuscript "Signatures of optimal codon usage predict metabolic ecology in budding yeasts" for consideration as a Research Article at PLOS Biology. Your manuscript has been evaluated by the PLOS Biology editors, by an Academic Editor with relevant expertise, and by four independent reviewers. Please accept my sincere apologies for the delay in sending the decision below to you.

In light of the reviews (below), we will not be able to accept the current version of the manuscript, but we would welcome re-submission of a much-revised version that takes into account the reviewers' comments. We cannot make any decision about publication until we have seen the revised manuscript and your response to the reviewers' comments. Your revised manuscript is also likely to be sent for further evaluation by the reviewers.

We expect to receive your revised manuscript within 3 months. 

**IMPORTANT - SUBMITTING YOUR REVISION**

Your revisions should address the specific points made by each reviewer. The Academic Editor has provided specific feedback, which I have included below, together with the reviewer comments, and which should be used as a guide during revision.

Please submit the following files along with your revised manuscript:

*Re-submission Checklist*

*Published Peer Review*

*PLOS Data Policy*

*Blot and Gel Data Policy*

Sincerely,

Gabriel Gasque, Ph.D.,

Senior Editor,

ggasque@plos.org,

PLOS Biology

REVIEWS:

Academic Editor:

Like the reviewers, I found the paper very interesting, but there is a bit of overselling of the implications. I recommend a major revision. Some of the recommendations by reviewers 3 and 4 are unrealistic as they would require extensive experimental work that would substantially delay any potential publication. Below are my reactions to reviewer comment and suggestions :

Rev #3: “Is GAL gene optimization not correlated with growth under other (non-galactose) growth conditions?”

This comment highlights that lack of certain control experiments. It is important to demonstrate that codon adaptation in GAL gene is specific to galactose growth. This requires measuring growth rate in other media in the laboratory.

Another important and feasible test is to demonstrate that codon adaptation of metabolic genes (not directly involved in galactose utilization) does not correlate with growth rate in galactose medium. This would require bioinformatic analysis only. This test is all the more important as certain species may be better able to optimize codon usage in all highly expressed genes due to differences in population size, regardless of the specific ecological conditions they have adapted to during evolution. 

Rev #3: “In S. cerevisiae, or other species in which this may have been explored, does increased GAL gene optimality (or increased genes expression) lead to better growth on galactose? I appreciate that insufficient expression would slow growth, but is more necessarily better? I suspect there is a point of diminishing returns or a point where 'enough' gene product is made for optimum growth. Does optimality drive the genes above that point? Would the genes still be there without the optimality?”

This is a lot of work, as it demands generating mutant strain with altered codon usage of the GAL genes. However, this highlights that the main conclusions on the predictive power of the method should be toned down.

I agree with reviewer 4 that “the presented results do not strictly assess how ecological variables can be predicted from the genomic pattern of codon usage. Such an approach would require at least some kind of linear modelling allowing to quantify the prediction error and some cross-validation approach to assess how accurate such prediction would be. “ Reviewer 2 also highlighted this problem: “But these hypotheses and predictions are actually not tested experimentally, leaving the validity of the approach (and of the initial hypothesis) not fully convincing.”

Therefore, the claims below should be toned down: 

“This work highlights the potential of codon optimization as a tool for gaining insights into the metabolic ecology of microbial eukaryotes. Doing so may be especially illuminating for studying fungal dark matter—species that have yet to be cultured in the lab or have only been identified by genomic material. “

Reviewer #1: Signatures of optimal codon usage predicts metabolic ecology in budding yeasts.

The submission from La Bella describes a detailed analysis of the association with optimal codon usage in the galactose metabolism pathway and ecological niche for 329 budding yeast species. The study exploits the diversity of whole genome sequences available for yeast species, many of which were originally obtaining by the authors' labs, and their earlier estimations of codon usage bias from 2019. There have been many studies of the evolution of the galactose gene cluster in yeasts, but this is one of the first (and certainly the most extensive) studies of codon adaptation, as a proxy for gene expression. The authors find that species with optimal codon usage in GAL genes (high expression) are associated with dairy environments (Saccharomycetaceae) or with humans (CUG-Ser1 clade). They also find a correlation between codon optimization in GAL genes and in thiamine biosynthesis genes, suggesting that there is a biochemical or ecological connection. The study is clear, well described and original, though the ability to predict metabolic ecology is a little overstated. For the CUG-Ser1 clade for example, it looks like it might be difficult to predict if newly identified species are associated with humans. The analysis is likely to be of general interest, including to yeast biologists, and metabolic ecologists.

Specific points.

1. Figure 1 is central to the analysis, but it is impossible to read in the printed version. Zooming in on the PDF does reveal the unfilled symbols, but they are impossible to see in a printout. It is also very difficult to see the variation in estA1 measurements using the color scale, and it is impossible to identify specific species without reference to earlier papers from the group. For example, I cannot easily pick out the human-enriched species with optimal codon usage referred to from lines 389. Some additional panels, or supplementary figures, would help (similar to Fig. 4). Fig 1B is too small relative to 1A. 

2. I have some problems understanding the correlation with the thiamine biosynthesis pathway, or more accurately, in understanding why K14154 was selected for discussion. As the authors note, this is number 8 in the list (Supplementary Table 3). Number 3 (K02564) is NAG1, a glucosamine-6-phosphate deaminase, involved in sugar (glucosamine) metabolism. Is it not more likely that correlation of galactose and glucosamine metabolism may be more important for ecology? NAG genes also tend to be clustered. Perhaps the authors have already examined codon optimization in NAG genes, which is worth a comment. Glucosamine metabolism may also be associated with first gene in the list (glycan synthesis). Number four, K01077 is PHO8 which also has role in thiamine synthesis, and could be added to Fig 5A. It would be useful to include KO definitions in Supplementary table 3.

Minor

3. Is the lack of GAL clusters/galactose metabolism in the Pichiaceae (spelled incorrectly in Fig. 1) and the Phaffomycetaceae discussed elsewhere?

4. The ecological enrichment (Fig. 3) is certainly interesting. Are the authors suggesting that changes in optimal codon usage occurred very recently (in evolutionary terms), because dairy associations are likely to be recent?

5. Page numbers are missing from references.

Reviewer #2: The manuscript by LaBella et al. aims at testing the hypothesis that codon optimality and gene function can be used to infer the ecology of yeasts. Such reverse ecology approaches are particularly interesting since ecological data show strong sampling -and reporting- bias. Codon optimilaty is an underused genomic signature which can be accessed in a relatively straightforward manner, given recent development in sequencing technologies. The approach is not whithout precedence: it was used very effectively, and experimentally validated, in bacteria to discover new genes involved in oxidative stress tolerance (doi: 10.1186/gb-2014-15-3-r44). It was also applied, and partly validated, on metagenomic data (https://doi.org/10.1093/nar/gkt673) and used recently on ecosystem data (doi: 10.7554/eLife.49816). Extending the approach to eukaryotic microbes is undoubtely promising.

Unfortunately, the study in its current state compares somewhat unfavorably to the works listed above in some aspects. Starting from the general hypothesis that signatures of codon optimality carry information on yeasts ecology, the authors provide a generally rigorous and well conducted analysis of codon optimality in GAL genes from the whole subphylum of budding yeasts. I very much appreciated that the authors used phylogenetic independent contrast to avoid lineage-dependent bias in their estimates of codon optimality. These analyses allow the authors to generate new, often experimentally tractable predictions (adaptation of Naumovozyma castellii and Kazachstania unispora to dairy environment, the association of thiamine metabolism with growth on galactose or in diary -but not alcohol- environment). But these hypotheses and predictions are actually not tested experimentally, leaving the validity of the approach (and of the initial hypothesis) not fully convincing. For this major reason, I feel like the current manuscript is too preliminary for publication, but represents a solid basis and promising lead. 

Major concerns:

RESULTS

1. The predictions generated by the analysis presented here are not validated experimentally. Therefore, there is no evidence to support the validity of the reverse ecology approach proposed. All predictions are probably not experimentally tractable, however some of the following should be tested:

. Deoptimize GAL genes or thiamine biosynthesis genes and check if there is a phenotypic effect specifically in the identified niche. 

. Test for the expression of GAL genes or accumulation of GAL proteins in some of the species and environment of interest

. L420 suggests that Spathaspora species would be "enable[d] or predispose[d them] to colonize human hosts". Any experiment to get support for this hypothesis?

. L452 "It is likely that [species] associated with both dairy and alcohol niches contain[s] species or populations that are better adapted to one niche than the other". Can't this be tested experimentally to validate the hypothesis?

2. "GAL codon optimization is correlated with growth rate on galactose". Correlation is not very convincing as it seems strongly driven by extreme values, especially for GAL7 where 2 species with low PIC are probably responsible for most of the correlation. In agreement, when CUG-Ser1 and sccharomycotina clades are considered separately, only GAL7 correlation remains significant in both clades. And even there, three species are probably strong drivers in the correlation observed for the two clades. The same concern applies to data presented in Figure 5B.

3. "GAL codon optimization is associated with specific ecological niches". The authors conclude that GAL genes optimization is higher in species from human-associated niches. Yet, codon optimization in species from human and/or insect niche is not significantly higher than that of species from neither human or insect niche. An alternative interpretation could be that association with insect decreases codon optimization. Can the authors exclude this alternative?

4. Have authors noted any relatioship between codon nitrogen content and ecological niche, as shown in doi: 10.1186/s13059-016-1087-9 ?

DISCUSSION

4. L566 "These results are especially promising as this method can be applied directly to genomic data—which is the only source of information for microbial dark matter known only from DNA". I feel like application may not be that straightforward as estimation of codon optimality often requires genome scale information, such as tRNA complement and codon adaptation values for all genes. Also ecological information can be deduced if optimilaty is high in genes with known function, which may not be very frequent. Can the authors discuss these possible limitations and how to circumvent them?

5. Can the authors discuss added value from their study in the context of previous codon optimality-based reverse ecology studies (eg doi: 10.1186/gb-2014-15-3-r44; https://doi.org/10.1093/nar/gkt673; doi: 10.7554/eLife.49816)?

Minor comments

The manuscript is very well written and very pleasant to read. Figures are of high quality. 

L498 "athways associated with galactose optimization" > "galactose metabolism optimization"

Reviewer #3: The authors explore the use of codon optimality as a tool for reverse ecology. The authors report positive correlations between GAL gene codon optimality and growth on galactose media. The authors show that yeasts that live in an environment with lactose have higher optimality of GAL genes. The authors also found a correlation between codon optimality in GAL and thiamine biosynthesis genes and then show that thiamine genes are also especially optimized in environments with lactose. 

Using codon optimality as a tool for reverse ecology is a very cool idea. It was, however, not clear to me from the paper how useful the tool is relative to other reverse ecology tools like gene presence or conservation. It would be helpful if the authors could highlight what discoveries this method allowed in this dataset, that would not have been found with other approaches. 

I was also confused about what species were included in the correlation analyses and how the growth rates were calculated. This confusion prevented me from getting too excited about the correlations presented, especially given the low strength of the correlations. For example, for the GAL growth/GAL optimal codon correlation, I couldn't shake the idea that the correlations could be driven by genes that were decaying and thus had low optimality and were also not able to foster growth on galactose media. I have listed questions/comments that arose as I read to make it clear areas where I required more information/explanation to fully understand.

Questions/suggestions:

In the abstract, I was confused by this sentence:

"For example, optimal codon usage of GAL genes is greater than 85% of all genes in the major human pathogen Candida albicans (CUG-Ser1 clade) and greater than 75% of genes in the dairy yeast Kluyveromyces lactis (family Saccharomycetaceae)." 

-I don't know what 85% of all genes means. Is it that the GAL genes use more optimal codons than 85% of all genes? I originally read it as 85% of codons are optimized. 

Is GAL gene optimization not correlated with growth under other (non-galactose) growth conditions? 

In the methods or elsewhere, I looked for an explanation of how codon optimality was determined for the different species. I found this:

"Codon optimization of individual GAL genes was assessed by calculating the species-specific tRNA adaptation index (stAI) from previously calculated species-specific codon relative adaptiveness (wi) value." This needs more explanation. Were there any empirical measurements that went into these values? If not, that should be stated explicitly and justified. 

In S. cerevisiae, or other species in which this may have been explored, does increased GAL gene optimality (or increased genes expression) lead to better growth on galactose? I appreciate that insufficient expression would slow growth, but is more necessarily better? I suspect there is a point of diminishing returns or a point where 'enough' gene product is made for optimum growth. Does optimality drive the genes above that point? Would the genes still be there without the optimality? 

"A total of 49 species' genomes had multiple copies of at least one GAL gene." Did that correlate with growth on Galactose? If not, maybe increased expression doesn't equate to optimal growth. Are the results the same without those 49?

Growth rate measurements require better explanation. "The growth rates for these species are 0.129, 0.339, and 0.211…" What do those rates mean? I am used to thinking of growth rate in terms of doubling time. How were the gal rates normalized with glucose rates?

"To determine if there is an association between codon optimization and the ability to grow on galactose, we compared optimization in the GAL pathway between species that are able and unable to grow on galactose. We found that species without evidence for growth on galactose had significantly lower (p < 0.05) codon optimization in GAL1 and GAL7 (Supplementary Figure 4)." Is this observed if one considers only genomes that have all the genes? If not, optimality is presented as a new way to find ecological info. Gene presence/absence is another way. I am curious if the optimality finds something gene presence/absence misses. Or, is the low optimality signal in the things not growing on glucose being dragged down by low optimality in a decaying pathway already missing components. 

Line 355 "We found a significant positive correlation between growth rate on galactose-

containing medium and codon optimization in the GAL pathway of genomes that have experienced translational selection on codon usage" Does this analysis and the others that follow exclude the species that do not grow on Gal? I know optimality is important, but low optimality seems unlikely to lead to no growth. But that no growth phenotype could be inflating the importance of optimality in the correlations. If those species were not excluded, I would like to see that analysis too. 

Line 378: There needs to be some more up-front explanation why the discussed niches might need different GAL gene expression. There are a lot of observations that are listed in this section, but I needed more explanation as to why those differences might be important to get the most out of reading the list. The Saccharomycetaceae analyses beginning on line 437 are more self-explanatory. 

What were the other things that were positively correlated with GAL gene optimality and why focus on the 8th in the list?

Reviewer #4: In this manuscript, LaBella and co-authors investigate how codon optimization (which can be computed from a complete genome sequence) can be used to predict ecological characteristics of budding yeasts. The underlying assumption is that the amount of codon usage optimization is a proxy for the level of gene expression, which in turns underlies the phenotype. More specifically, codon optimization allows pointing at highly expressed genes, and such genes are very likely to be of ecological importance. The authors illustrate this principle using three genes of the GAL pathway, and growth on Galactose as a resulting phenotype.

The authors convincingly demonstrate that different groups of yeast species with distinct ecologies display significant differences in codon optimization in the three GAL genes. The statistical analyses are very well conducted, and notably account for phylogenetic non-independence and multiple testing. While I generally agree with the detailed conclusions of the authors, I am more sceptical regarding the central claim of this manuscript, as reflected in the title. That different species with contrasting ecologies show significant differences in codon optimization does not necessarily implies that the latter allows to predict the former, as this intrinsically depends on the variance of the predictor (here codon usage optimization). As such, the presented results do not strictly assess how ecological variables can be predicted from the genomic pattern of codon usage. Such an approach would require at least some kind of linear modelling allowing to quantify the prediction error and some cross-validation approach to assess how accurate such prediction would be. My guess is (but I may, of course, be wrong) that this accuracy would be rather low, given the large variance in the correlation between codon usage and growth (Figure 2), as well as the overlap between the distributions of optimal codon usage between species groups (figure 3). On a side note, I would say that the results presented here, at best, show that growth on Galactose can be predicted from codon usage. While these results potentially can constitute a proof-of-concept for other studies, it might still be a strong take to claim that such methods can predict metabolic ecology in general.

Minor:

l181+: it would be useful to mention how tRNA quantities were estimated. Is it based on tRNA annotations in the genome (gene copy numbers), or from the sequencing of the tRNA pool in the cell?

l250: As far as I'm aware of, the ade4 package does not contain PIC of PGLS methods. Do the authors mean the 'ape' package?

l370: the PGM genes have not been introduced before. As I understand, these are used as a negative control... a sentence to clarify this would be helpful.

l490: randomly chosen VPS4 gene. A more rigorous approach would consist in defining a statistic based on the branch lengths, and compare the value of the GAL genes to the empirical distribution obtained by computing the statistic on a random sample of genes (rather than taking a single gene).

l512-514: "PIC correlation implies adaptation". I think this is a bit of a strong statement. This analysis shows that the correlation is not due to phylogeny only. But as the authors mention further in the manuscript, this could be due to a spurious correlation with another variable (for instance, shared metabolic intermediates). The PIC analyses do not preclude this possibility.

---

## [Decision Letter · Decision Letter 2]

12 Feb 2021

Dear Dr Rokas,

Thank you for submitting your revised Research Article entitled "Signatures of optimal codon usage associate metabolic function with ecology in budding yeasts" for publication in PLOS Biology. I have now obtained advice from the original reviewers and have discussed their comments with the Academic Editor. 

Based on the reviews, we will probably accept this manuscript for publication, provided you satisfactorily address the remaining points raised by the reviewers. Having discussed these with the Academic Editor, we don't think the experiments suggested by reviewer 2 are essential for publication in our journal, but we would welcome them if you wish to add them to strengthen the study. However, you should include the analysis suggested by reviewer 4, since it is straightforward.

We also want to suggest a change in your title, to "Signatures of optimal codon usage inform about the metabolic ecology of budding yeasts"

Please also make sure to address the data and other policy-related requests listed below my signature. 

We expect to receive your revised manuscript within two weeks. 

*Published Peer Review History*

*Early Version*

Sincerely,

Gabriel Gasque, Ph.D.,

Senior Editor,

ggasque@plos.org,

PLOS Biology

DATA POLICY:

We note that you have provided all your sequence analyses in Figshare repositories or in the supplementary files. In addition, we need you to provide all individual quantitative observations that underlie the data summarized in the figures and results of your paper. For an example see here: http://www.plosbiology.org/article/info%3Adoi%2F10.1371%2Fjournal.pbio.1001908#s5

These data can be made available in one of the following forms:

Regardless of the method selected, please ensure that you provide the individual numerical values that underlie the summary data displayed in the following figure panels: Figures 2A-C, 3AB, and 4BC.

Please also ensure that each figure legend in your manuscript includes information on where the underlying data can be found and that your supplemental data file/s has/have a legend.

Reviewer remarks:

Reviewer #1: The authors have made substantial efforts to address the concerns of the reviewers. The most important revisions include:

1. Changes to the title and some text that tones down the claims to predictive power

2.Including a description of the difference in growth rate on galactose and GAL codon optimization in S. cerevisiae and S. uvarum

3. Attempting to experimentally test the power of the predictive power using K. wickerhamii and W. occidentalis. It should be noted that this was not entirely successful. It might have been more convincing to test more species, or to test species with larger differences between the predicted the predicted growth rates.

4. Demonstrating that GAL gene optimization is not correlated with growth on mannose, and that codon optimization of a mannose metabolism gene PMI40 is not correlated with growth on galactose. The latter point is not particularly convincing however because codon optimization of PMI40 is not correlated with growth rate on mannose either, and was possibly not the best choice to test.

5. Including a more detailed description of genes in other KEGG pathways that correlated with optimization of GAL genes.

I do not have sufficient expertise to comment on how important the study is to the general area of ecological prediction. However, in my opinion the data is strong and the analysis is interesting.

Reviewer #2: I appreciated the additional analyses performed by the authors to adress previous comments on their manuscript. Although more limited than experimental testing, this provides support to the conclusions and interpretations of this work, and clarified several ambiguous points from the previous version. Following previous recommendations and comments, I will be very much looking forward to thorough experimental testing of this work in follow up studies. In the meantime, I think that two points still require attention in this revised version:

1.

"the correlation between GALactose metabolism genes and that growth rate on galactose is specific to this pathway" (L535) - Is this really the case? This seems to contradict the idea that "codon optimization is a valuable genomic feature for linking metabolic pathways with ecological niches" (L149). Is galactose a special case for which codon optimization correlates well with growth rate on the processed substrate? 

(L534) "This suggests not only that PMI40 is highly expressed regardless of metabolic state, but also that optimization in some pathways may better correlate with growth rate than in others". Could this hypothesis be tested quickly in a few selected conditions by Q-RT-PCR?

2.

I found the "Prediction of Growth Rate from Unsampled Genomes" section a valuable addition to the revised manuscript. "we generated 1,000 coding sequences with identical amino acid sequences to the observed GAL protein that had randomly assigned codon usage" (L378), to assess the predictive power of this analysis, it should be conducted on all genomes (codon optimized or not) for which growth rates on galactose are known. How often do efficient galactose users fall within the 95% CI? Reciprocally, how often do poor galactose users have predicted growth values outside the 95%CI? 

Reviewer #3: The authors have addressed my concerns.

Reviewer #4: The authors submitted an extensive revision of their manuscript, addressing the comments pointed by the editor. The revised manuscript includes (1) a rewriting of the text and title to better match the study's outcome and (2) new results strengthening the manuscript's central claim. I only have minor comments on the additional analyses, listed below.

1) l302+: It is unclear why the PGLS analysis did not need a new tree, compared to PICs. PICs can be seen as a particular case of PGLS, and both depend on an underlying phylogenetic tree. Can the author clarify this point?

2) Growth rate prediction: the authors run one model per gene and obtain three growth rate estimates, one per gal gene. Would it be possible (and make sense) to combine data of the three genes to get a single estimate? A confidence interval of the estimates could be provided, either by maximum likelihood from the PGLS prediction or by means of a bootstrap approach, resampling codon sites in all genes. 

3) Figure 4. B: for K. lactis, it is unclear where the 75% quantile line is. Maybe an arrow on the x-axis would be more precise if the dashed line is too close to the gene values? (alternatively, indicate the gene values with arrows and the quantiles with dashed lines.) Quantiles could also be displayed in subfigure 4C, for consistency.

---

## [Editor Report · Decision Letter 3]

15 Mar 2021

Dear Dr Rokas,

On behalf of my colleagues and the Academic Editor, Csaba Pál, I am pleased to say that we can in principle offer to publish your Research Article "Signatures of optimal codon usage in metabolic genes inform budding yeast ecology" in PLOS Biology, provided you address any remaining formatting and reporting issues. These will be detailed in an email that will follow this letter and that you will usually receive within 2-3 business days, during which time no action is required from you. Please note that we will not be able to formally accept your manuscript and schedule it for publication until you have made the required changes.

PRESS

Thank you again for supporting Open Access publishing. We look forward to publishing your paper in PLOS Biology. 

Sincerely, 

Gabriel Gasque, Ph.D. 

Senior Editor 

PLOS Biology